# Finally Outshining the Random Baseline:
# A Simple and Effective Solution for Active Learning
# in 3D Biomedical Imaging

**Carsten T. Lüth**[1,2,3*]**, Jeremias Traub**[1,2,4*]**, Kim-Celine Kahl**[1,2,3]**, Till Bungert**[1,2,3]**,
Lukas Klein**[1,2,5]**, Lars Kraemer**[1,2,3]**, Paul F. Jaeger**[2,6]**,
Klaus Maier-Hein**[1,2,3,7,8†]**, Fabian Isensee**[1,2,3†]

[1]*German Cancer Research Center (DKFZ) Heidelberg, Division of Medical Image Computing, Germany*
[2]*Helmholtz Imaging, German Cancer Research Center (DKFZ), Heidelberg, Germany*
[3]*Faculty of Mathematics and Computer Science, University of Heidelberg, Germany*
[4]*German Cancer Research Center (DKFZ) Heidelberg, Division of Intelligent Medical Systems, Germany*
[5]*Institute for Machine Learning, ETH Zürich, Switzerland*
[6]*German Cancer Research Center (DKFZ) Heidelberg, Interactive Machine Learning Group, Germany*
[7]*Pattern Analysis and Learning Group, Department of Radiation Oncology, Heidelberg University Hospital, Germany*
[8]*National Center for Tumor Diseases (NCT) Heidelberg, Germany*

*{carsten.lueth, jeremias.traub}@dkfz-heidelberg.de*
*\*/†: These authors contributed equally to this work.*

**Reviewed on OpenReview:** *https://openreview.net/forum?id=UamXueEaYW*

## Abstract

Active learning (AL) has the potential to drastically reduce annotation costs in 3D biomedical image segmentation, where expert labeling of volumetric data is both time-consuming and expensive. Yet, existing AL methods are unable to consistently outperform improved random sampling baselines adapted to 3D data, leaving the field without a reliable solution. We introduce Class-stratified Scheduled Power Predictive Entropy (ClaSP PE), a simple and effective query strategy that addresses two key limitations of standard uncertainty-based AL methods: class imbalance and redundancy in early selections. ClaSP PE combines class-stratified querying to ensure coverage of underrepresented structures and log-scale power noising with a decaying schedule to enforce query diversity in early-stage AL and encourage exploitation later. Our implementation within the nnActive framework queries 3D patches and uses nnU-Net as segmentation backbone. In our evaluation on 24 experimental settings using four 3D biomedical datasets within the comprehensive nnActive benchmark, ClaSP PE is the only method that generally outperforms improved random baselines in terms of both segmentation quality with statistically significant gains, whilst remaining annotation efficient. Furthermore, we explicitly simulate the real-world application by testing our method on four previously unseen datasets without manual adaptation, where all experiment parameters are set according to predefined guidelines. The results confirm that ClaSP PE robustly generalizes to novel tasks without requiring dataset-specific tuning. Within the nnActive framework, we present compelling evidence that an AL method can consistently outperform random baselines adapted to 3D segmentation, in terms of both performance and annotation efficiency in a realistic, close-to-production scenario. Our open-source implementation and clear deployment guidelines make it readily applicable in practice. Code is at `https://github.com/MIC-DKFZ/nnActive`.

# 1 Introduction

Annotation in 3D biomedical imaging is particularly expensive due to the requirement for highly specialized expertise and the inherently time-consuming nature of creating detailed segmentation masks for volumetric data (Litjens et al., 2017). Any approach that reliably reduces annotation effort in 3D biomedical imaging has the potential to unlock new tasks and applications for deep learning models in clinical and research settings where annotation cost represents the main bottleneck. Consequently, reducing the need for fully annotated datasets has become a major research focus. Various strategies are being explored, including enhanced annotation tools with interactive segmentation (Diaz-Pinto et al., 2024), improvements of the model training via self-supervised learning (Zhou et al., 2021; Wald et al., 2024), semi-supervised learning (Li et al., 2020), and learning from partial annotations (Can et al., 2018) or pretrained foundation models (Ma et al., 2024a). These approaches share the common goal of minimizing manual labor for annotation while maintaining or improving model performance.

Active Learning (AL) offers a promising strategy which is orthogonal to all the aforementioned approaches and aims to reduce annotation costs by selectively querying only the most informative data points for annotation, thereby maximizing model performance with minimal labeling effort. As the annotation cost reduction of AL upon application can not be validated (validation paradox) (Lüth et al., 2023) which hinders both method selection and optimization, it is of critical importance that an AL method demonstrates strong empirical evidence to yield reductions in annotation cost in a *realistic scenario* (Settles, 2011; Munjal et al., 2022).

*However, despite its transformative potential, the effectiveness of AL in reducing annotation costs remains largely unproven for 3D biomedical image segmentation.*

Several studies emphasize that random sampling remains a surprisingly strong baseline (Nath et al., 2021; Burmeister et al., 2022), and show that commonly used AL methods do not consistently outperform it (Gaillochet et al., 2023a;b; Vepa et al., 2024). Föllmer et al. (2024) state that 'Further research is necessary to prove the effectiveness of active learning for medical image segmentation'. Most notably, the only two works that rigorously evaluate random strategies specifically adapted to the 3D biomedical context (improved random strategies) report that, under current methodological standards, there is insufficient evidence to generally recommend AL over *improved random baselines* (Lüth et al., 2025; Burmeister et al., 2022), despite the naive random baselines being commonly outperformed.

Our proposed query method, Class-stratified Scheduled Power Predictive Entropy (**ClaSP PE**), is designed to be a generalizing solution to reduce annotation cost. It combines two simple yet effective extensions to a standard uncertainty-based AL method that directly addresses their empirically observed shortcomings in the context of 3D biomedical segmentation:

1. A stratification of standard uncertainty and class-specific uncertainties, which directly addresses the voxel-wise imbalance of classes while still retaining the ability to prioritize hard-to-predict cases.
2. An exponential scheduler for Power-Noising of scores (Kirsch et al., 2023) which addresses the low diversity of queries especially in early stage AL by perturbing the scores stronger in early AL stages and gradually reducing the noise towards later stages.

ClaSP PE is the first AL method for 3D biomedical image segmentation with compelling evidence to achieve general annotation cost reductions during application scenarios as it outperforms both standard and improved random baselines in terms of segmentation quality whilst not sacrificing annotation efficiency. We base this strong claim on the most comprehensive evaluation of AL methods for 3D biomedical segmentation to date which captures a wide range of realistic evaluation scenarios. We clarify our claim of realism for our evaluation based on the nnActive framework in section 2 alongside the challenges of applying AL to 3D biomedical segmentation.

The empirical evidence from our evaluation is delivered in two steps: As a first step, in section 4, we demonstrate that ClaSP PE consistently outperforms all other AL methods and random sampling strategies on the nnActive benchmark (Lüth et al., 2025), the most comprehensive benchmark to date for AL in 3D biomedical imaging. This encompasses four 3D biomedical datasets, each with three annotation budgets

(Label Regimes) that are evaluated with two distinct query designs (query patch sizes), resulting in 24 distinct experimental setups for AL experiments. In the second step, in section 5, we validate the generalization capabilities of ClaSP PE on four additional datasets by explicitly simulating real-world use-case scenarios (Roll-Out), demonstrating its practical applicability and robustness beyond the benchmark setting. We make sure to set up all parameters for the AL pipeline during Roll-Out according to our *Guidelines for Real-World Deployment* without manual adaptions which can serve as a recipe for practitioners when applying ClaSP PE to novel datasets and tasks.

In summary, our main contributions are:

- We propose ClaSP PE, a simple and effective query method that systematically addresses key limitations of current uncertainty-based AL methods.
- We conduct a large-scale evaluation, demonstrating that ClaSP PE brings reliable performance improvements over standard and improved random sampling baselines for 3D biomedical image segmentation on the nnActive benchmark spanning four datasets and six annotation budgets each.
- We provide evidence for the generalization capability of ClaSP PE by means of a Roll-Out study on four additional datasets to explicitly simulate a real-world use-case with all parameters being set based on our *Guidelines for Real-World Deployment*.

We wish to emphasize that the focal point of our work does not lie in methodological novelty but in providing a simple solution obtained by intuitive adaptations of existing methods for the challenging and long-standing problem of general effectiveness in 3D biomedical AL, which is backed up by empirical rigorous evaluation (Lipton & Steinhardt, 2019).

## 2 Challenges of Active Learning for 3D Biomedical Image Segmentation

The design and evaluation of AL pipelines must account for the characteristics of 3D biomedical segmentation, or it risks not delivering on its promise of reducing annotation effort. We will now start by giving a short recap on segmentation for 3D biomedical images and then introduce our approach for evaluation followed by highlighting the key differentiating factor to previous works in AL for 3D biomedical segmentation which is the query design as a 3D patch.

**Segmentation on 3D biomedical images.** 3D volumetric images are very large, often exceeding $500 \times 500 \times 500$ voxels for a single volume (e.g., an upper body CT scan). These images oftentimes feature many homogeneous structures, such as organs, which are located in specific characteristic areas of the images. Further, these datasets commonly contain a dominant background class that occupies most of the volume but is not a target of interest, and there frequently exist strong volumetric differences between different structures or classes of interest, such as most tumors being much smaller than organs. The community for 3D biomedical images has adapted to these challenges by designing specific training techniques where less frequent classes are oversampled and models are either trained on smaller 3D patches of the data or 2D slices (Isensee et al., 2021; 2024) with 3D U-Net-like models (Ronneberger et al., 2015) generally performing best.

**Evaluation of Active Learning Methods.** Our evaluation directly builds upon Lüth et al. (2025), who propose the nnActive framework and benchmark, which directly address four pitfalls commonly occurring in the evaluation of AL in 3D biomedical imaging.[1] Concretely, these pitfalls are (1) Evaluation is restricted to too few settings; (2) Model Training does not incorporate partial annotations; (3) Random Baseline is not adapted to 3D setting; (4) Annotation cost is measured in voxels. The occurrence of these pitfalls directly hinders the ability to draw conclusions regarding the reduction of annotation effort in practically relevant settings. The framework and benchmark address these by: (1) ensuring a diverse set of datasets and multiple annotation budgets (Label Regimes); (2) using nnU-Net with partial loss, ensuring well-configured models that make efficient use of annotations during training; (3) comparing our AL method against *improved random baselines* (Foreground Aware Random strategies) which oversample foreground regions in a class-balanced fashion to handle the inherent class imbalance between foreground and background as well as between different foreground classes; (4) proposing the Foreground Efficiency (FG-Eff) measure which relates

---

[1]For detailed information, we refer to this paper.

the number of queried foreground voxels to the model performance by means of an exponential fit, we can identify whether an AL method selects foreground more effectively rather than just selecting more of it. The exact details of our evaluation are given in section 4.

**Query Design.** The nnActive Framework combines multiple improvements over the evaluation schemes of related works and most notably uses 3D nnU-Net with partial loss (Isensee et al., 2021; Gotkowski et al., 2024) which enables arbitrary design of a query (e.g. 3D patches, 2D slices or single voxels). The general design of the query is a crucial factor in AL for 3D segmentation, requiring a careful trade-off between allowing the human to annotate queries efficiently whilst allowing the Query Method (QM) to focally query structures of interest. When annotating entire 3D images, a lot of effort is spent annotating regions with redundant information which is why it is typically better to use partial annotations in form of 2D slices or 3D patches, especially when the used AL method can find the most informative regions.

We utilize 3D query patches of fixed size in combination with a partial loss integrated into nnU-Net (Gotkowski et al., 2024; Isensee et al., 2021), allowing us to train 3D models following Lüth et al. (2025). This design strikes a balance between annotation efficiency and informativeness while maintaining flexibility in query selection, as the query patch size can be selected based on the structures of interest instead of model constraints. The combination of 3D query design and 3D models represents a major differentiating factor of our work from most related works, which either rely on querying entire 3D images (Nath et al., 2021), or restrict queries to 2D slices with 2D models (Burmeister et al., 2022; Gaillochet et al., 2023a;b; Ma et al., 2024b; Föllmer et al., 2024; Vepa et al., 2024; Shi et al., 2024a).

While the ability of a QM to directly select 3D patches corresponding to regions of interest is elegant and potentially powerful, it also introduces significant complexity to the general query algorithm with multiple overlapping candidate patches. This complexity largely hinders the implementation of representation-based QMs, such as Core-Set (Sener & Savarese, 2018), or more sophisticated uncertainty-based QMs like USIM (Föllmer et al., 2024), due to both runtime and memory constraints arising from the transition from 2D slices to 3D patches[2]. As our input shape is not necessarily the query patch shape, it is an open research question what a representation of our query patch is. Generally, obtaining representations for 3D volumes is a major challenge for AL as noted by Liu et al. (2023) in their evaluation for starting budget selection. Further, there is a general consensus that even for 2D slices and 2D models, representation-based methods like Core-Set are performing worse than uncertainty-based AL methods (Burmeister et al., 2022; Föllmer et al., 2024). We hypothesize that this stems from the skip connections of the utilized U-Nets (Ronneberger et al., 2015), which may lead to the representations, typically taken from the bottleneck layers, not capturing the fine details necessary to allow optimal data selection.

## 3    Method

Our proposed query strategy, Class-stratified Scheduled-Power Predictive Entropy (ClaSP PE), is designed to improve AL for 3D biomedical segmentation by effectively balancing informativeness, class representation, and diversity of the queried patches and thereby solves prominent issues of top-k sampling uncertainty methods (as illustrated in fig. 1). Starting from a standard **Uncertainty-Based scoring** commonly employed in top-k sampling which returns an uncertainty map $u(x)$ for each image $x$, we introduce two key modifications: Class Stratified Sampling and an Exponential Scheduler for Score Perturbation. Importantly, these extensions are agnostic to the specific uncertainty scoring function used and can be applied on top of any existing uncertainty-based method.

**Class Stratified Sampling.** To encourage class-balanced selection of queries, we implement a stratified sampling procedure. Specifically, we select an equal number of patches per predicted class based on the model's predictions. For each image $x$, we compute class-specific uncertainty scores

$$u_c(x) = p_c(x) \cdot u(x), \tag{1}$$

---

[2]For example, on the KiTS dataset, one median 3D volume has $\sim 188 \times 10^3$ potential queries using patches compared to $\sim 500$ queries using slices using the setup described in appendix C

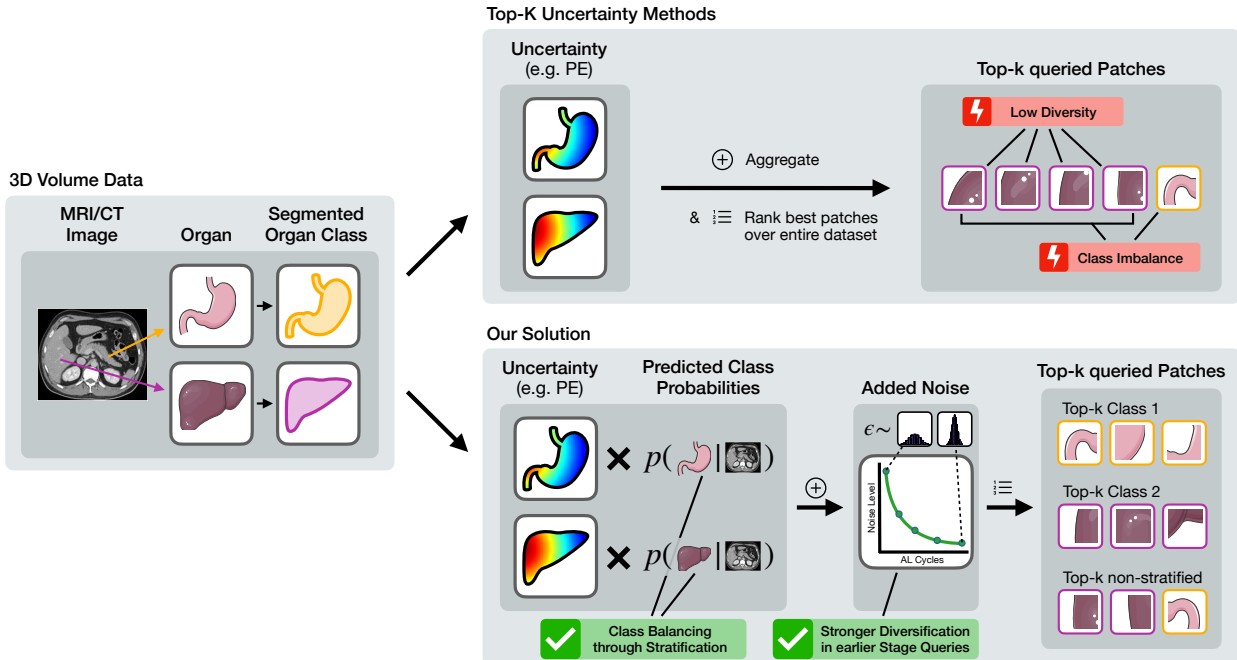

Figure 1: **Overview of the ClaSP PE query strategy.** We overcome two key limitations of standard uncertainty-based Active Learning methods (e.g. Predictive Entropy), class imbalance and low diversity of the queries, by adding two simple modifications: (1) class-stratified sampling for 66% of the query budget based on predicted class probabilities, and (2) a scheduler decreasing the noise for score perturbation via log-scale power noising to enhance diversity during query selection.

where $p_c(x) = p(Y = c|x)$ denotes the predicted probability for class $c$. Patches are then ranked per class according to $u_c(x)$, and the top $N_c$ patches from each class are selected, where $N_c$ is chosen such that all classes contribute equally to the stratified subset. This ensures that underrepresented classes are not neglected, which naturally supports metrics that average performance across classes (e.g., mean Dice). Importantly, by leveraging the model predictions our approach does not require any additional label information. To our knowledge, balancing queries in this way has not been used in the AL literature before. Crucially, only a fraction $\alpha$ of the samples is selected using this stratified approach, with the remaining $1 - \alpha$ samples being selected based on the standard uncertainty map $u(x)$ to retain sensitivity to highly uncertain examples regardless of class distribution.

**An Exponential Scheduler for Score Perturbation via Log-scale Power Noising.** To enforce diversity among selected queries, especially in earlier AL cycles, we apply power noising to the scores (on patch-level) before selecting the top-k samples (Kirsch et al., 2023). Specifically, we perturb the scores on a logarithmic scale by adding Gumbel noise $\epsilon \sim \text{Gumbel}(0, \beta^{-1})$. Additionally, we use an exponential schedule[3] for the perturbation strength $\beta^{-1}$ such that it decreases towards later AL cycles from an initial value $\beta_0^{-1}$ to a final value $\beta_{\max}^{-1}$, in order to gradually shift the focus from exploration to exploitation:

$$\beta(t) = \exp\Big(\big[1 - \tfrac{t}{T}\big]\ln(\beta_0) + \tfrac{t}{T}\ln(\beta_{\max})\Big), \quad t = 0, \ldots, T \tag{2}$$

where $t$ indexes the current AL cycle and $T$ is the total number of AL cycles.

For our final ClaSP PE method we utilize Predictive Entropy to obtain uncertainty-based scores as it was highlighted as the overall best performing AL method on the nnActive benchmark (Lüth et al., 2025). We then apply the stratified selection to $\alpha = 66\%$ of the budget based on our analysis in section 4.2. For the

---

[3]We also experimented with linear and sigmoid schedules but found that exponential schedules generally performs on par or better.

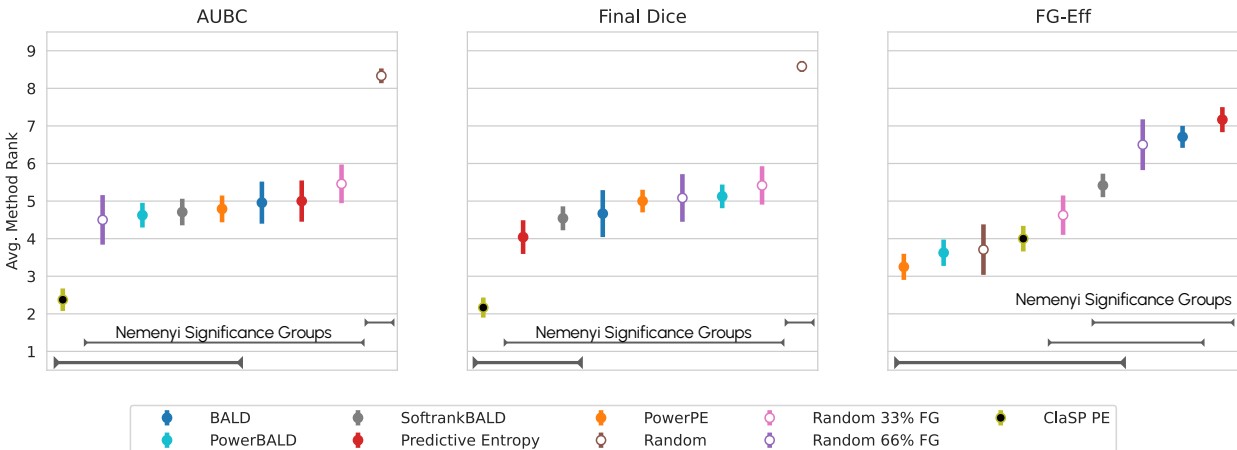

Figure 2: **ClaSP PE delivers substantial performance improvements without sacrificing annotation efficiency.** The plots show average method rankings (lower is better) with standard error for AUBC, Final Dice, and FG-Eff across the nnActive benchmark. Results are aggregated over 4 datasets, 3 Label Regimes, and 2 query patch sizes, each evaluated with 4 random seeds, providing robust estimates of method performance. The brackets indicate groups of methods that do not differ significantly based on a post-hoc Nemenyi test at significance level 0.05.

exponential scheduler, we fixed $\beta_0 = 1$ and $\beta_{\max} = 100$ for all evaluation settings and no additional tuning was performed.

This method is simple to implement and flexible, yet effective, as our empirical studies in sections 4 and 5 demonstrate. We provide an implementation of ClaSP PE in the nnActive framework (Lüth et al., 2025) and a detailed pseudo-code of the method in appendix B.

## 4 Experimental Results on the nnActive Benchmark

We evaluate the effectiveness of our proposed query strategy ClaSP PE on the nnActive benchmark (Lüth et al., 2025), which is, to our knowledge, the most comprehensive AL suite currently available for 3D biomedical segmentation. To this end, we perform over 1000 nnU-Net training runs across 24 distinct settings (4 datasets × 3 Label Regimes × 2 query patch sizes) including dedicated ablations. This comprehensive setup captures a wide range of segmentation challenges and enables statistically meaningful conclusions about the robustness, efficiency, and generalizability of our method.

**Datasets, Label Regimes & query patch sizes.** The nnActive benchmark spans four prominent medical imaging datasets: AMOS2022 (challenge task 2) (Ji et al., 2022), Medical Segmentation Decathlon–Hippocampus (Antonelli et al., 2022), KiTS2021 (Heller et al., 2023), and ACDC (Bernard et al., 2018). Each of these datasets is evaluated under three distinct Label Regimes (Low-, Medium- and High-Label) corresponding to a specific annotation budget defined as a number of total patches. Further, the entire benchmark entails two distinct query patch sizes (referred to as Main and Patch$\times\frac{1}{2}$), with the latter being half the size along each dimension. For more information regarding datasets and Label Regimes, we refer to appendix C.

**Baselines.** We compare ClaSP PE against the standard random baseline and two improved random baselines (Random 33% and 66% FG) (Lüth et al., 2025), as well as the following five uncertainty-based QMs: Predictive Entropy (Settles, 2009), Bayesian Active Learning by Disagreement (BALD) (Houlsby et al., 2011; Gal et al., 2017), PowerBALD (Kirsch et al., 2023), SoftrankBALD (Kirsch et al., 2023), and PowerPE (Kirsch et al., 2023). Random 33% and 66% FG simulate the process of selecting a patch around a random foreground region for $X\%$ of their budget. See appendix D for more details.

**Experimental Setup.** Our experimental setup is identical to the nnActive benchmark using four seeds with a fixed test split, and using a custom nnU-Net trainer with 200 Epochs in the 3D full resolution configuration with each AL experiment consisting of 5 cycles. We evaluate AL performance with the following metrics operating on the mean Dice score (Dice, 1945): The Final Dice score achieved after the final AL cycle; the Area Under Budget Curve (AUBC) (Zhan et al., 2021; 2022) which aggregates the mean Dice scores across one AL trajectory over all cycles to measure the overall performance; the Foreground Efficiency (FG-Eff) (Lüth et al., 2025), which acts as a proxy for annotation efficiency by setting the performance in relation to the queried foreground voxels by means of an exponential fit; the Pairwise Penalty Matrix (PPM) (Ash et al., 2020), which quantifies along the entire AL trajectory how often one method significantly outperforms another based on paired t-tests [4], and can thus simply be aggregated over e.g. datasets. The exact implementation and more details with regard to the evaluation metrics are provided in appendix D.

**Results.** As our baseline models are well adapted to medical datasets by means of proper Data Augmentation, Model Architecture and loss formulation, we observe as expected that absolute performance gains for single datasets can be small in absolute value (Mittal et al., 2019; Lüth et al., 2023; Beck et al., 2021). Therefore, our evaluation is performed on the highest aggregation level as the goal of AL is to bring generalizing performance improvements for a specific annotation budget. Figure 2 shows the method rankings averaged across the nnActive benchmark. Exact numerical results are provided in appendix E. We find that ClaSP PE achieves the best overall performance in terms of both AUBC and Final Dice, generally outperforming both improved random baselines and established AL methods. Importantly, our approach delivers these segmentation quality gains while maintaining high annotation efficiency, as indicated by FG-Eff: although ClaSP PE does not always achieve top FG-Eff, it consistently ranks among the most efficient methods. This reflects an inherent interplay between segmentation performance and annotation efficiency, where methods that strongly focus on highly informative regions can improve Dice scores but may risk inefficient use of annotated foreground (e.g., Predictive Entropy). Our ablations (see section 4.2) further show that score perturbation is crucial for preventing such inefficiencies, and that gradually reducing the noising strength boosts segmentation performance at the cost of only a slight reduction in FG-Eff. Overall, ClaSP PE achieves a favorable balance across this trade-off, providing efficient, informative, and diverse query selection through our proposed modifications.

In addition to the average rankings, fig. 2 includes statistical significance groups derived from the conservative Nemenyi post-hoc test (Nemenyi, 1963) with a significance level of $p = 0.05$. These groups provide exploratory evidence for the robustness of ClaSP PE: it forms a distinct top-performing group for segmentation performance measured by AUBC and Final Dice, while also remaining competitive in FG-Eff. In contrast, the naive random baseline is consistently ranked lowest and is significantly outperformed by all other methods. Overall, ClaSP PE shows the most consistent separation from random and uncertainty-based baselines across all three metrics. Importantly, although SoftrankBALD also appears in the top Nemenyi group, ClaSP PE shows a clearer overall advantage when considering both the average rankings (fig. 2) and absolute performance (table 1). Detailed results of the Nemenyi tests are provided in appendix E.1.

Additionally, when comparing the average Final Dice and AUBC over all settings, ClaSP PE is the only AL method that improves over improved random strategies, as shown in table 1. Both PowerBALD and PowerPE outperform their top-k counterparts BALD and Predictive Entropy for the Final Dice performance metric contrary to the rankings in fig. 2 which provides further evidence for the more stable performance of these methods across annotation budgets, as already noted in Lüth et al. (2025).

ClaSP PE performs well overall and generally delivers substantial performance improvements on the KiTS dataset, as can be seen in table 6 and table 7. However, especially on the AMOS dataset for smaller annotation budgets, ClaSP PE underperforms improved random strategies, but shows smaller underperformance compared to the other AL methods (shown in table 7). This behavior is further discussed in section 4.1.

For ACDC and Hippocampus, the absolute performance differences are generally small (table 6) and often fall within the respective error bars. This highlights two important points: (1) broad evaluation across many datasets and label regimes is essential to reveal overall trends, and (2) even when such trends clearly

---

[4] These are performed without family-wise error rate correction following (Ash et al., 2020; Beck et al., 2021; Föllmer et al., 2024)

Table 1: **ClaSP PE achieves better average performance than both random and AL baselines.** Average Performance aggregated over all 24 distinct AL settings of the nnActive benchmark for AUBC and Final Dice alongside the 95% Confidence Interval (higher is better as indicated by green colorization). Details for the computation are given in appendix E.7.

| Query Method | AUBC | Final Dice |
|---|---|---|
| BALD | $62.39 \pm 0.30$ | $65.43 \pm 0.41$ |
| PowerBALD | $64.81 \pm 0.35$ | $67.93 \pm 0.29$ |
| SoftrankBALD | $63.74 \pm 0.32$ | $67.32 \pm 0.28$ |
| Predictive Entropy | $63.27 \pm 0.40$ | $67.35 \pm 0.58$ |
| PowerPE | $64.85 \pm 0.35$ | $68.01 \pm 0.38$ |
| Random | $60.57 \pm 0.39$ | $61.65 \pm 0.43$ |
| Random 33% FG | $66.00 \pm 0.27$ | $69.74 \pm 0.32$ |
| Random 66% FG | $67.14 \pm 0.22$ | $71.14 \pm 0.22$ |
| ClaSP PE | $67.62 \pm 0.33$ | $72.81 \pm 0.30$ |

Figure 3: **ClaSP PE consistently outperforms both random and AL baselines across the nnActive benchmark.** The Pairwise Penalty Matrix summarizes statistically significant wins and losses from pairwise t-tests (p=0.05) between methods. Results are aggregated over 24 distinct AL settings on the nnActive benchmark, including 4 datasets × 3 Label Regimes × 2 query patch sizes. Remaining lose scenarios against Random 66% FG stem from challenging Low-Label settings on the AMOS dataset (discussed in section 4.1).

favor a given method, this does not imply that it will yield significant gains over all other methods in every individual scenario.

To complement the aggregate metric rankings and average segmentation performance, fig. 3 presents the PPM, assessing pairwise performance differences on the nnActive benchmark. ClaSP PE clearly emerges as the strongest method overall, outperforming all random and AL baselines more frequently than it is outperformed. This underscores the method's robustness and generalizability across diverse settings. Further, we show that the overall trends of the PPM are persistent across different p-values and when using the Bonferroni-Holm method (Holm, 1979) to account for the family-wise error rate appendix E.6. Nonetheless, in roughly 20% of the comparisons, Random 66% FG surpasses ClaSP PE. These cases are concentrated almost exclusively on the AMOS dataset under Low-Label Regimes, a particularly challenging scenario due to the high number of classes and the constrained annotation budget. We investigate this dataset-specific behavior in more detail in section 4.1.

Finally, we note that the combination of score perturbation and stratified sampling substantially boosts the performance of standard Predictive Entropy across all evaluation metrics. Our large-scale evaluation provides

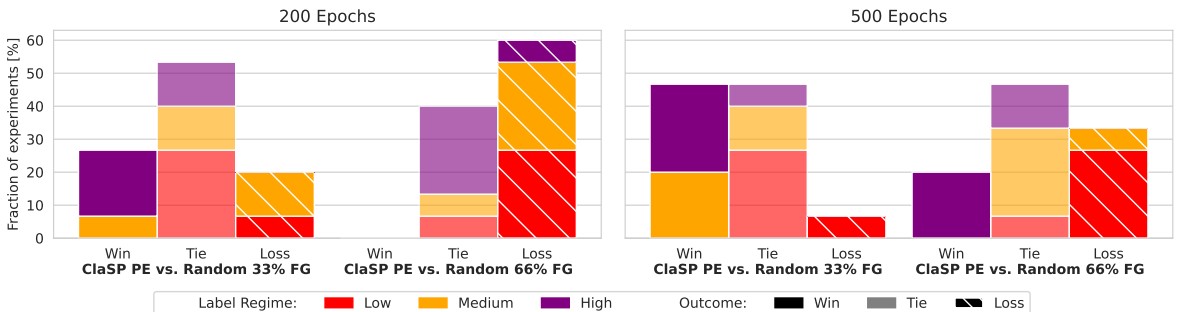

Figure 4: **Longer training amplifies the advantage of ClaSP PE over random selection.** Shown are fractions of significant wins, losses, and resulting ties of ClaSP PE against improved random baselines on the AMOS dataset, as computed via the PPM. We compare models trained for 200 (left) and 500 (right) epochs, as well as different Label Regimes (color-coded). Each Label Regime carries 33% of the entire fraction of experiments which is then divided into wins, losses and ties. While at 200 epochs ClaSP PE loses on 60% of the experiments to FG66 and ties in the rest, it outperforms Random FG 66% in 20%, ties in 48% and loses in only 32% when trained for 500 epochs.

clear empirical evidence for the effectiveness and robustness of these simple yet impactful modifications. Additional qualitative analyses can be found in appendix I.

## 4.1 Investigating Loss Scenarios on AMOS

To better understand the limited performance gains of ClaSP PE compared to improved random baselines on the AMOS dataset, we conducted an ablation study that evaluates the influence of longer training on AL performance.

Specifically, we compare the performance of ClaSP PE against the improved random baselines (Random 33% FG and Random 66% FG) on the Low-, Medium-, and High-Label Regimes (with a total budget of 200, 1000, and 2500 patches, respectively). All methods are trained for 200 and 500 epochs, and we conduct the comparison on the Main nnActive Benchmark, which results in 3 distinct evaluation settings.

We observe that increasing the training duration from 200 to 500 epochs substantially improves the win-to-lose ratio of ClaSP PE relative to the random baselines. Figure 4 shows that in the 500-epoch setting, the number of lose-cases is reduced and primarily confined to the lower Label Regimes. In particular, ClaSP PE now consistently outperforms Random 66% FG in the High-Label Regime, whereas the Low-Label Regime is still dominated by lose-cases. Compared to the Random 33% FG baseline, ClaSP PE shows clear and consistent gains in both the Medium- and High-Label Regimes, underscoring the benefits of extended training. Detailed results are shown in appendix E.4.

These findings suggest that longer training amplifies the advantage of ClaSP PE over random selection. We hypothesize that the large number of 15 classes on AMOS makes the Low-Label especially challenging as the 200 patches annotation budget, when evenly spaced across all classes, could capture less than 14 examples per class (compared to 67 on KiTS, for 3 classes). This highlights the sensitivity of AL performance not only to the training dynamics but also to task-specific factors such as the number of classes. Further, we observe in an analysis for AMOS with class-level dice that the loss scenarios on the low-label regime mainly stem from the segmentation performance on the right and left adrenal gland which is also less frequently queried compared to Random 66%FG. We show the detail in appendix E.5 We therefore emphasize the importance of adapting the annotation budget to the number of classes for practitioners.

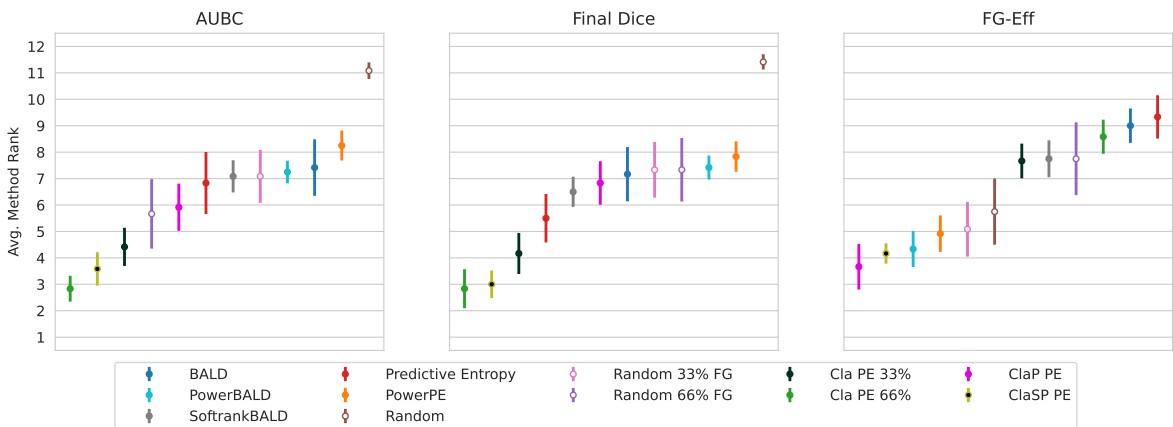

Figure 5: **ClaSP PE achieves the best trade-off between segmentation quality and annotation efficiency.** Average method rankings on the nnActive Main benchmark (4 datasets × 3 Label Regimes × 1 query patch size), with additional method variants, Cla PE 66%, Cla PE 33% and ClaP PE.

## 4.2 Ablating the Influence of ClaSP PE Components

Our proposed method, ClaSP PE, combines two simple yet effective components: (1) class-balanced sampling applied to a certain fraction of queries, and (2) log-scale power noising applied to the scores prior to top-k patch selection. In this ablation, we analyze the contribution of each component and justify our final design choice. To this end, we evaluate additional method variants, *Cla PE* with $\alpha = 33\%$ and $\alpha = 66\%$ to isolate the effect of class-balanced sampling without power noising and further ablate the fraction of queries for which it is applied, as well as *ClaP PE* which is identical to ClaSP PE using $\alpha = 66\%$ but uses a constant noise value $\beta = 1$ instead of a scheduler. We report their performance across the nnActive Main benchmark.

From the aggregated results, displayed in fig. 5, we observe the following: (1) Class-balanced querying improves performance across the board: Both Cla PE 66% and Cla PE 33% outperform standard PE on all evaluation metrics. Moreover, higher stratification rates lead to better segmentation quality: We find that increasing the fraction of stratified queries from 33% to 66% yields improvements in AUBC and Final Dice, with only a minor decrease in FG-Eff. (2) The addition of power-noising substantially improves the FG-Eff, indicating improved annotation cost-efficiency through enhanced diversity, but leads to a reduction in absolute performance measured by AUBC and Final DICE, as can be observed when comparing ClaP PE and Cla PE 66%. (3) Gradually decayed power noising leads to the overall best tradeoff with regard to annotation efficiency and absolute performance as it is across all three metrics among the best. This supports the notion that the decaying schedule leads to a more diverse set of queries in early iterations of AL, which gradually become more focused on harder cases when the model has adapted to the data distribution. Detailed results are shown in appendix E.2.

Overall, the combination of 66% stratified querying and gradually decayed power noising provides the best trade-off between segmentation quality and annotation efficiency, justifying the choice of ClaSP PE as our final method.

## 5 Simulating Real-World Active Learning in a Roll-Out Study

To evaluate the generalization and practical utility of ClaSP PE, we conduct a roll-out study across a diverse set of real-world biomedical segmentation datasets. Importantly, we do not perform any dataset-specific finetuning, treating this as a plug-and-play scenario that mirrors how one might apply ClaSP PE in practical, previously unseen tasks.

The methods we compare include our proposed ClaSP PE, standard Predictive Entropy, which ranked just behind ClaSP PE on the nnActive benchmark, uniform random sampling, and Random 66% FG, a stronger baseline incorporating foreground-aware sampling.

We follow all design decisions of the nnActive experiment setup, such as the starting budget and dataset preprocessing, but introduce two new components tailored for real-world deployment: (1) a **systematic selection of query patch size** based on the median connected component sizes of the target structures, and (2) **normalized query budgets**, set to 50 or 100 patches per class depending on task complexity (e.g. the expected homogeneity). These additions ensure that queries remain representative and task-appropriate. Our full Guidelines for Real-World Deployment are provided in appendix F.

We evaluate performance on four datasets that vary widely in task complexity, number of foreground classes, and annotation difficulty: LiTS (Bilic et al., 2023), a two-class foreground segmentation task for liver and tumor; WORD (Luo et al., 2022), a 16-class organ segmentation task; Tooth Fairy 2 (Bolelli et al., 2025; 2024; Lumetti et al., 2024), which requires dense labeling of 42 dental structures; and MAMA MIA (Garrucho et al., 2025), a lesion segmentation task with a single target class. A fixed data split is used for all experiments (75% train & pool, 25% test), which is identical across four random seeds. Detailed dataset characteristics are provided in appendix C.

As summarized in Table 2, ClaSP PE overall performs on par or better than all baseline methods across datasets and metrics. It delivers reliable segmentation quality improvements while maintaining or exceeding annotation efficiency, without any task-specific method tuning. While Random shows high FG-Eff on LiTS and WORD, this results from querying only a very small amount of foreground, which artificially inflates FG-Eff without translating into segmentation performance gains. Predictive Entropy partially shows competitive performance with ClaSP PE in terms of segmentation performance, while ClasP PE demonstrates improved FG-Eff over PE across all roll-out datasets. On the large scale MAMA MIA breast cancer dataset, featuring many redundant structured for a highly complex task, ClaSP PE performs substantially better. Further, the results on the nnActive benchmark (fig. 2) reveal that PE fails to reliably outperform random baselines, whereas ClaSP PE shows consistent improvements. Together, these results underscore the robust out-of-the-box performance of the ClaSP PE method and establish it as a practical and effective solution for active learning in real-world 3D biomedical segmentation tasks.

Similarly, the PPM shown in fig. 6 reveals that ClaSP PE showcases the overall best performance being never significantly outperformed by Random and Random 66% FG while winning in over 50% of all cases and also outperforming Predictive Entropy significantly in 25% of all cases while being significantly outperformed in 5%. We provide detailed results in appendix G.

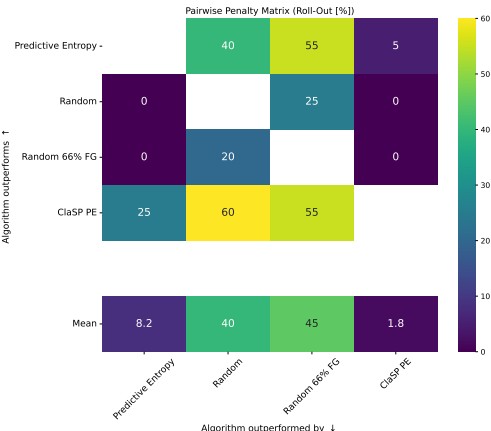

Figure 6: **ClaSP PE shows overall strongest performance on the roll-out study.** PPM for the roll-out study aggregated over all settings. In all settings, ClaSP PE wins against or ties with the random baselines.

Table 2: **ClaSP PE provides robust performance gains on out-of-the-box deployment.** Performance on the Roll-Out datasets, measured by AUBC, Final Dice, and FG-Eff (higher is better, indicated by green colorization).

| Dataset ($n_{samples}$) | LiTS (n=99) | | | WORD (n=90) | | | Tooth Fairy 2 (n=360) | | | MAMA MIA (n=1130) | | |
|---|---|---|---|---|---|---|---|---|---|---|---|---|
| Metric | AUBC | Final Dice | FG-Eff | AUBC | Final Dice | FG-Eff | AUBC | Final Dice | FG-Eff | AUBC | Final Dice | FG-Eff |
| Random | 51.23 | 52.38 | 46.25 | 77.35 | 78.03 | 3.66 | 61.83 | 64.32 | 11.88 | 55.23 | 58.24 | 39.13 |
| Random 66% FG | 48.63 | 50.05 | 1.27 | 78.19 | 78.25 | 1.34 | 65.30 | 68.61 | 10.85 | 44.38 | 45.10 | -4.67 |
| Predictive Entropy | 57.81 | 65.38 | 38.94 | 78.43 | 78.96 | 0.91 | 66.65 | 71.97 | 16.25 | 59.07 | 64.74 | 9.43 |
| ClaSP PE | 60.30 | 65.80 | 39.60 | 78.27 | 78.42 | 1.33 | 67.32 | 71.49 | 20.07 | 63.85 | 68.62 | 57.36 |
| 100% Data Dice | | 77.3 | | | 80.7 | | | 72.6 | | | 71.0 | |

## 6 Limitations

While ClaSP PE demonstrates strong performance across both benchmark and roll-out evaluations, several limitations remain. First, like all AL methods, it faces the risk of benchmark-specific overfitting, due to the necessity of empirically validating design decisions (Shi et al., 2024a; Föllmer et al., 2024; Gaillochet et al., 2023b; Vepa et al., 2024). Our dual evaluation mitigates this concern but cannot fully eliminate it. Further, as the entire evaluation is based on the average Dice which is the default overlap-based metric for semantic segmentation (Maier-Hein et al., 2024), our results do not necessarily extend to boundary-based evaluation metrics or when only specific classes are of interest.Second, the method depends on the predictive capacity of the underlying model: when initial segmentation quality is insufficient, stratified querying becomes less effective, though our guidelines for employing ClaSP PE mitigate this risk, and the use of pre-trained models may further improve early-stage segmentation quality (Gupte et al., 2024). Third, AL is inherently an economic trade-off: reduced annotation cost must be weighed against additional computational overhead, and the optimal balance is context dependent (Settles, 2011). Fourth, while we compared against established strong baselines, more complex AL strategies (s.a. Hübotter et al. (2024); Föllmer et al. (2024)) could potentially offer further gains, though their adaptability for querying 3D patches remains uncertain. Fifth, ClaSP PE relies on a small set of hyperparameters governing stratification and power-noising. Although validated across diverse datasets, these may benefit from adaptive tuning to better match dataset-specific characteristics. Finally, since our empirical evidence is obtained using the nnActive framework with 3D patches as query design, conclusions may differ under meaningful deviations from it, such as alternative segmentation backbones (Munjal et al., 2022) or 2D slice queries. A detailed discussion of these limitations is provided in Appendix H.

**On the Importance of Query Design and Annotation Technique.** The design of the query, whether it is a whole 3D image, a 3D volumetric patch, a 2D slice, or even a single voxel, substantially impacts the annotation process and tooling efficiency. However, no consensus exists on which query design and annotation process, such as sparse annotation, super-pixels/voxels, or scribbles, is the most economical, as each one has its own advantages and drawbacks depending on the specific task and currently available tooling (Tajbakhsh et al., 2020; Shi et al., 2024b). We consider annotation technique selection critical for maximizing economic effectiveness.

Our evaluation uses 3D patches, which support various annotation processes including sparse 2D slice-wise schemes (Çiçek et al., 2016; Burmeister et al., 2022) and scribble annotations (Li et al., 2024; Gotkowski et al., 2024). With promptable foundation models like SAM (Kirillov et al., 2023), MedSAM (Ma et al., 2024a), and nnInteractive (Isensee et al., 2025), 3D patches as annotation tools 3D patches enable targeted annotation, verification, and correction of specific structures within localized image regions. We focused on selecting informative patches rather than explicitly evaluating these annotation processes; examining how different techniques interact with patch-based querying remains future work.

# 7 Conclusion

We propose ClaSP PE, the first AL query method with substantial evidence of reducing annotation effort over random strategies for 3D biomedical segmentation in a close-to-production environment. ClaSP PE offers consistent performance gains across a wide range of datasets and AL scenarios. In addition to its strong performance, ClaSP PE is conceptually lightweight and easy to implement, enabling seamless integration into existing AL frameworks. Its computational cost remains comparable to standard top-k selection methods, making it well-suited for practical deployment.

**For developers and researchers**, ClaSP PE can serve as a strong and easy-to-implement baseline for future AL research. Our open-source code and results reduce the experimental overhead for developers and enable fair and reproducible comparisons in methodological studies.

**For practitioners**, our implementation of ClaSP PE offers a solution that can be integrated into real-world annotation workflows. It comes embedded in an AL pipeline that includes guidelines for setting all relevant parameters. This allows it to be implemented efficiently for use in the 3D biomedical segmentation domain when used inside the nnActive framework. For real-world deployment, the results of our evaluation lead to the following recommendations:

- Use ClaSP PE within the nnActive framework querying 3D patches, using the auto-configuration of nnU-Net.
- Train models for 1000 epochs, as AL performance generally improves for longer training durations.
- Follow our Guidelines for Real-World Deployment for patch size and query size (see appendix F).

## Acknowledgments

This work was funded by Helmholtz Imaging (HI), a platform of the Helmholtz Incubator on Information and Data Science. This work is supported by the Helmholtz Association Initiative and Networking Fund under the Helmholtz AI platform grant (ALEGRA (ZT-I-PF-5-121)).

The authors gratefully acknowledge the computing time provided on the high-performance computer HoreKa by the National High-Performance Computing Center at KIT (NHR@KIT). This center is jointly supported by the Federal Ministry of Education and Research and the Ministry of Science, Research and the Arts of Baden-Württemberg, as part of the National High-Performance Computing (NHR) joint funding program (https://www.nhr-verein.de/en/our-partners). HoreKa is partly funded by the German Research Foundation (DFG).

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

## Author Contributions

This work was carried out over 6 months and the core idea for the algorithm was developed by Fabian Isensee, Carsten Lüth, and Jeremias Traub. The exact implementation of the algorithm was done by Carsten Lüth, and then Jeremias Traub and Carsten Lüth designed the experiments and ablated the design decisions. All experiments were performed by Jeremias Traub. The writing was done by Jeremias Traub and Carsten Lüth in equal parts with reviews by all other authors.

# Appendix

## Table of Contents

## A   Task Description

As we use the AL framework proposed by Lüth et al. (2025), we refer to their work for a detailed task (Appendix B). Here, we only provide a high-level overview.

In the context of Active Learning (AL) for 3D biomedical image segmentation, acquiring complete annotations for entire volumetric scans is often prohibitively expensive and time-consuming, due to the need for expert annotators and the high dimensionality of the data. To address this, recent approaches advocate for the use of partial annotations, where only selected subregions of a 3D image–such as spatial patches–are labeled. This strategy enables models to learn effectively while significantly reducing annotation effort. The AL process is thus centered around a query method that strategically selects informative regions to annotate, allowing training to proceed using only a subset of the full data.

This framework can be formalized by considering the training data as 3D volumetric images $X \in \mathbb{R}^{M \times H \times W \times D}$ with dense labels $Y \in \{1, \ldots, C\}^{H \times W \times D}$. Rather than providing the full Y, a query function Q(X) identifies subsets $\tilde{Y} \subseteq Y$ for annotation. Specifically, this work focuses on querying 3D patches within each image, defined by locations and patch sizes. During training, only the labeled regions $\tilde{Y}$ are used to compute the loss, with the unannotated portions ignored or treated with weak supervision. This partial supervision setup allows the AL framework to scale efficiently to large 3D datasets without the prohibitive cost of full annotation.

## B   ClaSP PE Algorithm

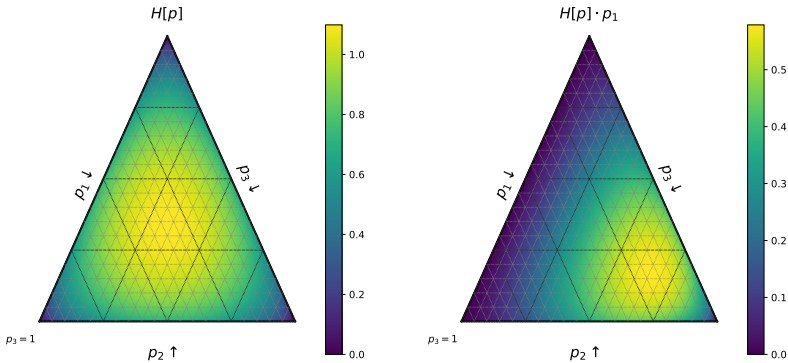

Figure 7: Ternary plot visualizing the difference of the entropy $u = H[p]$ and our proposed class-specific measure $u_1 = H[p] \cdot p_1$ for $y \in \{1, 2, 3\}$.

We start by giving a short recap of our proposed query method (QM) to introduce the notation. Followed by additional implementation details to support reproducibility by means of two complementary representations of the algorithm for ClaSP PE.

**Class Stratified Sampling**   Given an image $x$, an uncertainty map $u(x)$, and predicted class probabilities $p_c(x) = p(Y = c|x)$, we obtain the class-specific scores

$$u_c(x) = p_c(x) \cdot u(x) \tag{3}$$

A direct example of how these class specific scores behave in a class scenarios is visualized in fig. 7. We then select samples in a stratified fashion for each class $c$ based on $u_c$, respectively. To our knowledge, this approach of balancing the queries using stratification has not been used in the AL literature before. Crucially, we do not select all samples with the stratified approach but only a fraction $\alpha$ with the remaining $1 - \alpha$ samples being selected based on the standard uncertainty map $u(x)$ to retain sensitivity to highly uncertain examples regardless of class distribution.

**An Exponential Scheduler for Score Perturbation via Log-scale Power Noising**   Our exponential scheduled power-noising is a straight extension of the work by Kirsch et al. (2023) works as follows:

$$s_{\text{ClaSP PE}}(t) = \log s_{\text{Cla PE}} + \epsilon(t) \tag{4}$$

where

$$\epsilon(t) \sim \text{Gumbel}(0, \beta^{-1}(t)) \tag{5}$$

with $t \in \{0, ..., T\}$ which represents the current AL cycle where $T$ is the maximum number of AL cycles counting only those with a Query step. $\beta_0$ is the initial value, while $\beta_{\max}$ is the final value for the last cycle.

$$\beta(t) = \exp([1 - \tfrac{t}{T}] \ln(\beta_0) + \tfrac{t}{T} \ln(\beta_{\max})) \tag{6}$$

**Implementations**   First, we provide a Python-style pseudocode in algorithm 1 that abstracts away specific implementation details, focusing instead on the core structure and logic of the method. Second, we present a fully detailed algorithmic version that outlines our exact implementation inside the nnActive framework shown in algorithm 2. This combination provides a high-level overview while also being transparent about our implementation.

As the high-level Python-style pseudocode abstracts away the patches, it therefore can serve as foundation for implementations where overlap checks are not necessary.

---

**Algorithm 1** Abstracted ClaSP PE in a Python-style pseudocode with patches abstracted away

---

**Input:** unlabeled_pool: unlabeled dataset, model: python model, t: current loop, T: max loop with query, beta_0: starting beta, beta_max: final beta, alpha: fraction stratified, num_classes: number of classes, n: query size

**PseudoCode**

```
u_images = []
for x in unlabeled_pool: # Computing ClaSP PE for a sample
    p = model.forward(x)
    u = entropy(p)
    u_c = cat(p[without bg_class] * unsqueeze(u, 0), unsqueeze(u, 0))
    u_c += gumbel_noise(u_c.shape, exp(-(1-t/T)*ln(beta_0) + t/T *ln(beta_max)))
    u_images.append(u_c)

# Selecting Query over entire samples s_budgets = floor(n*alpha/C)
query = [] for c in range(C[without bg_class]):
    best = argsort(u_images[:, c])
    best.pop(i) for i in query
    query.append(best[::-1][s_budgets])
best = argsort(u_images[:, c])
best.pop(i) for i in query
query.append(best[::-1][1- (s_budgets)*C))
return query
```

---

---

**Algorithm 2** Exact ClaSP PE algorithm as implemented in the nnActive Framework

---

**Input:**
Set of images $\{X^{(i)}\}_{i=1}^{N}$, query size $n$, labeled set $\mathcal{L}$, Uncertainty function $\mathcal{U}$, number of classes $C$, fraction class specific $\alpha$, aggregation method with scheduled powernoiseing ($A$)

**Output:** Final query set $\mathcal{Q}$

1: $\tilde{\mathcal{Q}} \leftarrow \{\emptyset\}_{c=1}^{C+1}$ # Initialize stratified query set
2: **for** each image $X^{(i)} \in \{X^{(i)}\}_{i=1}^{N}$ **do**
3:      $P \leftarrow \mathcal{M}(X)$ # compute probability for image
4:      $U \leftarrow U(X^{(i)}, \mathcal{M})$ # compute uncertainty for image
5:      $U_{\text{Agg}} \leftarrow A(\mathcal{U})$ # aggregate uncertainties to patch-level
6:      $Q_{\text{Image}} \leftarrow \{\emptyset\}_{c=1}^{C+1}$ # initialize best patches for current image
7:      **for** $c \in \text{Shuffle}(\{1, ..., C\})$ **do**
8:          $U_c \leftarrow U \cdot P_c$
9:          $U_{c,\text{Agg}} \leftarrow A(\mathcal{U})$ # aggregate uncertainties to patch-level
10:          **for** $q$ in $\text{sort}(U_{c,\text{Agg}})[::\text{-}1]$ **do** # sort in descending order according to uncertainty
11:              **if** $\text{overlap}(q, \mathcal{Q}_{\text{Image}} \cup \mathcal{L}) \leq o$ **then** # ensure that
12:                  $\mathcal{Q}_{c,\text{Image}} \leftarrow \mathcal{Q}_{c,\text{Image}} \cup \{q\}$
13:              **end if**
14:              **if** $\text{len}(\mathcal{Q}_{c,\text{Image}}) >= \alpha * n/C$ **then**
15:                  Break
16:              **end if**
17:          **end for**
18:      **end for**
19:      **for** $q$ in $\text{sort}(U_{\text{Agg}})[::\text{-}1]$ **do** # sort in descending order according to uncertainty
20:          **if** $\text{overlap}(q, \mathcal{Q}_{\text{Image}} \cup \mathcal{L}) \leq o$ **then** # ensure that
21:              $\mathcal{Q}_{C+1,\text{Image}} \leftarrow \mathcal{Q}_{C+1,\text{Image}} \cup \{q\}$
22:          **end if**
23:          **if** $\text{len}(\mathcal{Q}_{C+1,\text{Image}}) >= \alpha * n/C$ **then**
24:              Break
25:          **end if**
26:      **end for**
27:      $\tilde{\mathcal{Q}} \leftarrow \mathcal{Q} \cup \mathcal{Q}_{\text{Image}}$
28: **end for**
29: **for** $c \in \{1, ..., C\}$ # Build final query with stratified samples **do**
30:      $Q \leftarrow Q \cup \text{sort}(\tilde{Q}_c)[:: -1][: \alpha * n/C]$
31: **end for**
32: $Q \leftarrow Q \cup \text{sort}(\tilde{Q}_c)[:: -1][: n - (\alpha * n/C)]$ # Add unstratified samples
33: **Return** $\mathcal{Q}$

---

# C   Dataset Details

Key characteristics of the datasets used in the nnActive benchmark (section 4) directly match with Lüth et al. (2025) and are shown in table 3. For the roll-out study (section 5), dataset characteristics are shown in table 4. All images are resampled to the median dataset spacing. Further details on the different segmentation tasks are given in table 5.

The MAMA MIA dataset is additionally preprocessed using only the subtraction image where the pre-contrast image is subtracted from the first available post-contrast image.

Table 3: Dataset details and configurations for the nnActive study.

| Dataset | ACDC | AMOS | KiTS | Hippocampus |
|---|---|---|---|---|
| # Classes w.o. Background | 3 | 15 | 3 | 2 |
| Median Shape | 16.5x237x206 | 237.5x582x582 | 526x512x512 | 36x50x35 |
| Used Spacing | 2x0.6875x0.6875 | 5x1.5625x1.5625 | 0.78125x0.78125x0.78125 | 1x1x1 |
| # Pool & Training | 150 | 150 | 225 | 195 |
| # Validation | 50 | 50 | 75 | 65 |
| Query Patch Size | 4x40x40 | 32x74x74 | 60x64x64 | 20x20x20 |
| Budget: Low [# Patches](% Voxels) | 150 (0.75%) | 200 (0.26%) | 200 (0.16%) | 100 (6,51%) |
| Budget: Medium [# Patches](% Voxels) | 300(1.50%) | 1000 (1.30%) | 1000 (0.80%) | 200 (13,02%) |
| Budget: High [# Patches](% Voxels) | 450(2.25%) | 2500 (3.25%) | 2500 (2.00%) | 300 (19,54%) |
| Query Patch Size (Patch$\times\frac{1}{2}$) | 2x20x20 | 16x37x37 | 30x32x32 | 10x10x10 |
| Budget: Low [# Patches](% Voxels) | 150 (0.09%) | 200 (0.03%) | 200 (0.02%) | 100 (0.77%) |
| Budget: Medium [# Patches](% Voxels) | 300(0.19%) | 1000 (0.16%) | 1000 (0.10%) | 200 (1.63%) |
| Budget: High [# Patches](% Voxels) | 450(0.28%) | 2500 (0.41%) | 2500 (0.25%) | 300 (2.44%) |
| Test set Mean Dice (1000 Epochs) | 0.912 | 0.893 | 0.773 | 0.895 |
| Test set Mean Dice (500 Epochs) | 0.912 | 0.883 | 0.751 | 0.895 |
| Test set Mean Dice (200 Epochs) | 0.910 | 0.860 | 0.705 | 0.895 |

Table 4: Dataset details and configurations for the roll-out study.

| Dataset | LiTS | WORD | Tooth Fairy 2 | MAMA MIA |
|---|---|---|---|---|
| # Classes w.o. Background | 2 | 16 | 42 | 1 |
| Median Shape | 495×512×512 | 200×512×512 | 169×344×371 | 80×256×256 |
| Used Spacing | 1×0.7676×0.7676 | 3×0.9766×0.9766 | 0.3×0.3×0.3 | 2×0.7031×0.7031 |
| # Pool & Training | 99 | 90 | 360 | 1130 |
| # Validation | 32 | 30 | 120 | 376 |
| Budget [# Patches](% Voxels) | 750 (0.19%) | 4,000 (15.8%) | 10,500 (4.5%) | 500 (0.09%) |
| Query Patch Size | 28×44×39 | 29×74×87 | 33×34×35 | 16×48×57 |
| Test set Mean Dice (1000 Epochs) | 0.799 | 0.845 | 0.752 | 0.765 |
| Test set Mean Dice (500 Epochs) | 0.797 | 0.829 | 0.745 | 0.746 |
| Test set Mean Dice (200 Epochs) | 0.773 | 0.807 | 0.726 | 0.710 |

# D   Active Learning Framework

Our work builds directly on the existing nnActive framework (Lüth et al., 2025), preserving its design choices to ensure seamless applicability in both benchmarking and real-world annotation workflows. To maintain compatibility with the nnU-Net training and data management pipeline, all annotation updates are performed within the nnU-Net dataset structure. In particular, we store all queried patch metadata in *loop_XXX.json* files within the *nnUNet_raw* folder, where each file corresponds to a particular AL loop and contains information about the queried regions. These modifications in the *nnUNet_raw* directory are automatically reflected in the preprocessed datasets used for training by running the standard *nnUNet_preprocessing* step. For the query stage, we follow the patch-wise inference strategy of nnU-Net.

Table 5: Foreground class names for all datasets.

| Dataset | Class names in order of labels (ascending) |
|---|---|
| ACDC | right ventricle, myocardium, left ventricular cavity |
| AMOS | spleen, right kidney, left kidney, gall bladder, esophagus, liver, stomach, aorta, postcava, pancreas, right adrenal gland, left adrenal gland, duodenum, bladder, prostate/uterus |
| Hippocampus | anterior hippocampus, posterior hippocampus |
| KiTS | kidney, kidney-tumor, kidney-cyst |
| LiTS | liver, cancer |
| WORD | liver, spleen, left_kidney, right_kidney, stomach, gallbladder, esophagus, pancreas, duodenum, colon, intestine, adrenal, rectum, bladder, Head_of_femur_L, Head_of_femur_R |
| Tooth Fairy 2 | Lower Jawbone, Upper Jawbone, Left Inferior Alveolar Canal, Right Inferior Alveolar Canal, Left Maxillary Sinus, Right Maxillary Sinus, Pharynx, Bridge, Crown, Implant, Upper Right Central Incisor, Upper Right Lateral Incisor, Upper Right Canine, Upper Right First Premolar, Upper Right Second Premolar, Upper Right First Molar, Upper Right Second Molar, Upper Right Third Molar (Wisdom Tooth), Upper Left Central Incisor, Upper Left Lateral Incisor, Upper Left Canine, Upper Left First Premolar, Upper Left Second Premolar, Upper Left First Molar, Upper Left Second Molar, Upper Left Third Molar (Wisdom Tooth), Lower Left Central Incisor, Lower Left Lateral Incisor, Lower Left Canine, Lower Left First Premolar, Lower Left Second Premolar, Lower Left First Molar, Lower Left Second Molar, Lower Left Third Molar (Wisdom Tooth), Lower Right Central Incisor, Lower Right Lateral Incisor, Lower Right Canine, Lower Right First Premolar, Lower Right Second Premolar, Lower Right First Molar, Lower Right Second Molar, Lower Right Third Molar (Wisdom Tooth) |
| MAMA MIA | lesion |

After all ensemble members have predicted each image, the AL method is applied in a final step to compute uncertainty maps and select patches to be labeled. Our implementation of standard top-k uncertainty-based methods, such as PE or BALD, follows the algorithm described in algorithm 3.

---

**Algorithm 3** Active Learning Patch Selection

---

**Input:**
Set of images $\{X^{(i)}\}_{i=1}^{N}$, query size $n$, labeled set $\mathcal{L}$, Uncertainty function $U$, Aggregation function $A$, $o$ allowed overlap **Output:** Final query set $\mathcal{Q}$

1: Initialize final query set $\mathcal{Q} \leftarrow \emptyset$
2: **for** each image $X^{(i)} \in \{X^{(i)}\}_{i=1}^{N}$ **do**
3:     $\mathcal{U} \leftarrow U(X^{(i)}, \mathcal{M})$ # compute uncertainty for image
4:     $\mathcal{U}_{\text{Agg}} \leftarrow A(\mathcal{U})$ # aggregate uncertainties to patch-level
5:     $\mathcal{Q}_{\text{Image}} \leftarrow \emptyset$ # initialize best patches for current image
6:     **for** $q$ in $\text{sort}(\mathcal{U}_{\text{Agg}})[::\text{-1}]$ **do** # sort in descending order according to uncertainty
7:         **if** $\text{overlap}(q, \mathcal{Q}_{\text{Image}} \cup \mathcal{L}) \leq o$ **then** # ensure that
8:             $\mathcal{Q}_{\text{Image}} \leftarrow \mathcal{Q}_{\text{Image}} \cup \{q\}$
9:         **end if**
10:     **end for**
11:     $\mathcal{Q} \leftarrow \mathcal{Q} \cup \mathcal{Q}_{\text{Image}}$
12: **end for**
13: $\mathcal{Q} \leftarrow \text{sort}(\mathcal{Q})[::\text{-1}]$ # sort in descending according to uncertainty
14: **Return** $\mathcal{Q}$

---

### D.1 Evaluation Metrics

We adopt the comprehensive set of evaluation metrics used in the nnActive benchmark (Lüth et al., 2025) to assess the performance of different QMs.

**Final Dice** The Final Dice score reflects the segmentation performance after the full annotation budget has been spent. It particularly emphasizes the effectiveness of a QM in the later stages of AL and allows for straightforward interpretation.

**Area Under the Budget Curve (AUBC)** The AUBC measures overall performance across the entire AL trajectory. It is computed as the area under the Mean Dice curve using the trapezoid method. Higher values indicate better performance. We normalize AUBC such that it lies in the range $[0, 1]$. We refer to Zhan et al. (2021; 2022) for further details.

**Pairwise Penalty Matrix (PPM)** The PPM compares methods pairwise using a two-sided t-test with significance level $\alpha = 0.05$ (see (Ash et al., 2020) for further details). It quantifies how often one method significantly outperforms another across datasets and Label Regimes. Each row shows the fraction of wins, and each column shows the fraction of losses, expressed in percentages.

**Foreground Efficiency (FG-Eff)** We use FG-Eff as a metric for annotation efficiency, quantifying how quickly a method reaches full-data performance as a function of the annotated foreground voxels (a proxy for annotation effort). FG-Eff is based on fitting an exponential decay curve:

$$y(t) = (\hat{y}(\hat{t}_0) - \hat{y}_{\text{full}}) \exp(-\gamma(t - \hat{t}_0)) + \hat{y}_{\text{full}} \tag{7}$$

Here, $t \in [0, 1]$ is the fraction of annotated foreground voxels, $\hat{y}_{\text{full}}$ is the model's Dice score on the full dataset, and $\hat{y}(\hat{t}_0)$ is its performance on the starting budget. A higher $\gamma$ (FG-Eff) indicates faster convergence to full performance with less annotation.

FG-Eff complements performance metrics by quantifying annotation efficiency. A good QM performs well in terms of FG-Eff and traditional metrics (Final Dice, AUBC, PPM). High FG-Eff with low overall performance

should be viewed skeptically, as the metric can be *hacked* by querying a very small amount of foreground. Importantly, FG-Eff is only meaningful when QMs are compared under the same model, training regime, and annotation budgets, since $\hat{y}_{\text{full}}$ and $\hat{y}(\hat{t}_0)$ are experiment-dependent. We refer to Lüth et al. (2025) for further details.

## D.2 Experiment Details

For the AL experimental setup, we follow Lüth et al. (2025): We use a starting budget and query size equal to 20% of the full annotation budget of each Label Regime. To ensure a representative starting budget, it is allocated to sample random foreground regions of each class, so that all classes are present in at least two patches. The rest of the starting budget is selected using the Random 33% FG strategy. Details on the annotation budget and query design for each nnActive benchmark dataset are provided in table 3. For the roll-out datasets (table 4), we employ the guidelines detailed in appendix F.

We use nnU-Net (Isensee et al., 2021), a self-configuring deep learning framework, as our segmentation model. If not explicitly stated otherwise, all models are trained for 200 epochs using the `3D full resolution` configuration of nnU-Net. To increase model robustness, we use an ensemble of five models trained via 5-fold cross-validation. We perform complete retraining of the models for each AL loop. The training of the models themselves is not seeded, but all dataset-related parameters are. All results are averaged across four seeds.

**Hyperparameters** We directly took the Random FG configurations from nnActive (Lüth et al., 2025). As standard $\beta$ values for PowerBALD, PowerPE and SoftrankBALD we used 1 as detailed in Kirsch et al. (2023) and following the evaluation in Lüth et al. (2025). For $\alpha$, the fraction of samples that is selected using the stratified approach in ClaSP PE, we compared 33% and 66% as shown in our ablations, following the same values as the FG percentage of the Random FG methods. The initial and final noising strength, $\beta_0 = 1$ and $\beta_{max} = 100$ were chosen following the evaluation of Lüth et al. (2025) (Appendix G.3), which parsed a similar range showing that the most crucial factor is a general reduction of $\beta$ for larger annotation budgets, and no further tuning of the method parameters was done.

For the Tooth Fairy 2 dataset, we train without mirroring. For runtime savings, we omit Test-Time Augmentation during validation for MAMA MIA and Tooth Fairy 2 and we set pred_tile_step_size = 0.75 for inference on MAMA MIA.

**Compute Resources** All experiments are performed as single-GPU trainings on A100 GPUs. In total, the large-scale evaluation of the ClaSP PE method on the nnActive benchmark and the roll-out datasets required around 20,000 GPU hours, each with around 180 GB of RAM.

# E    nnActive Benchmark Results

In this section, we provide detailed results on the nnActive benchmark. We refer to the *nnActive main benchmark* as the experiment configuration described in Section 5.1 in Lüth et al. (2025), which encompasses 12 distinct settings across 4 datasets and 3 Label Regimes. Further extending the method evaluation, Lüth et al. (2025) define a Patch$\times\frac{1}{2}$ setting, which uses a query patch size that is halved along each dimension compared to that of the main benchmark. The specific settings are provided in table 3.

## E.1    Results aggregated over Main Benchmark and Patch$\times\frac{1}{2}$ Setting

The results presented in this section are aggregated over both the main benchmark and the Patch$\times\frac{1}{2}$ setting, resulting in 24 distinct experiment configurations across 4 datasets, 3 Label Regimes, and 2 query patch sizes. Specifically, fig. 8 shows the results of Nemenyi post-hoc tests, based on Friedman tests (Demšar, 2006), to analyze the significance of performance differences, and fig. 9 shows the PPMs for each dataset.

The Friedman tests are conducted across all $k = 9$ methods under comparison using $N = 24$ configurations (i.e., 24 paired performance outcomes per method) and show significant results (at $p = 0.05$). The Nemenyi post-hoc analysis evaluates all pairwise differences in average ranks. Using the standardized z-score of a Nemenyi test with ranking difference $\Delta$ (Demšar, 2006)

$$z = \Delta \left/ \sqrt{\frac{k(k+1)}{6N}} \right. \tag{8}$$

we can compute the effect size as $r = z/\sqrt{N}$. In our setup, following Cohen's guidelines (Cohen, 1988), the effect sizes of small ($r = 0.1$), medium ($r = 0.3$), and large ($r = 0.5$) correspond to the average ranking differences of $\Delta \approx 0.39$, $\Delta \approx 1.16$, and $\Delta \approx 1.94$, respectively. As an example, an average ranking difference of 0.5 would correspond to a small effect size of around 0.129, which highlights the conservative nature of the Nemenyi test, particularly with many methods and a relatively small sample size (Nemenyi, 1963), meaning that some practically meaningful differences may not reach significance.

We report the exact p-values in fig. 8 and use a significance threshold of 0.05 to form the groups shown in fig. 2. The resulting significance groups should be interpreted as exploratory evidence rather than definitive proof of method superiority; indeed, the Nemenyi test is conservative, which means that the significant separations observed in our results likely understate, rather than overstate, the true differences between methods (Nemenyi, 1963).

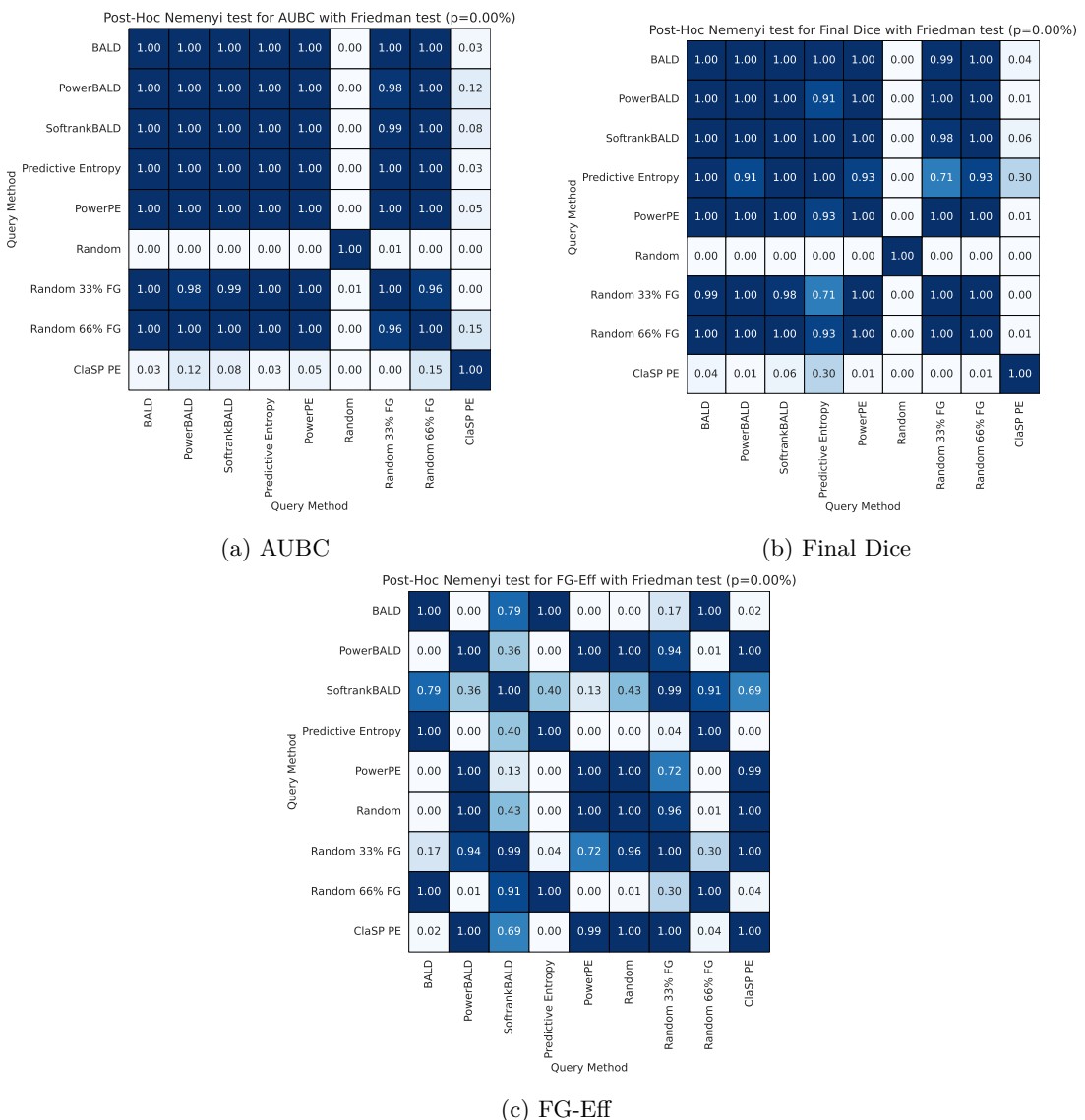

(a) AUBC

(b) Final Dice

(c) FG-Eff

Figure 8: p-values for the Nemenyi post-hoc tests, based on Friedman tests, on the nnActive benchmark for all evaluation metrics. Results are aggregated across 4 datasets × 3 Label Regimes × 2 query patch sizes. The corresponding significance groups for $p = 0.1$ are indicated in fig. 2.

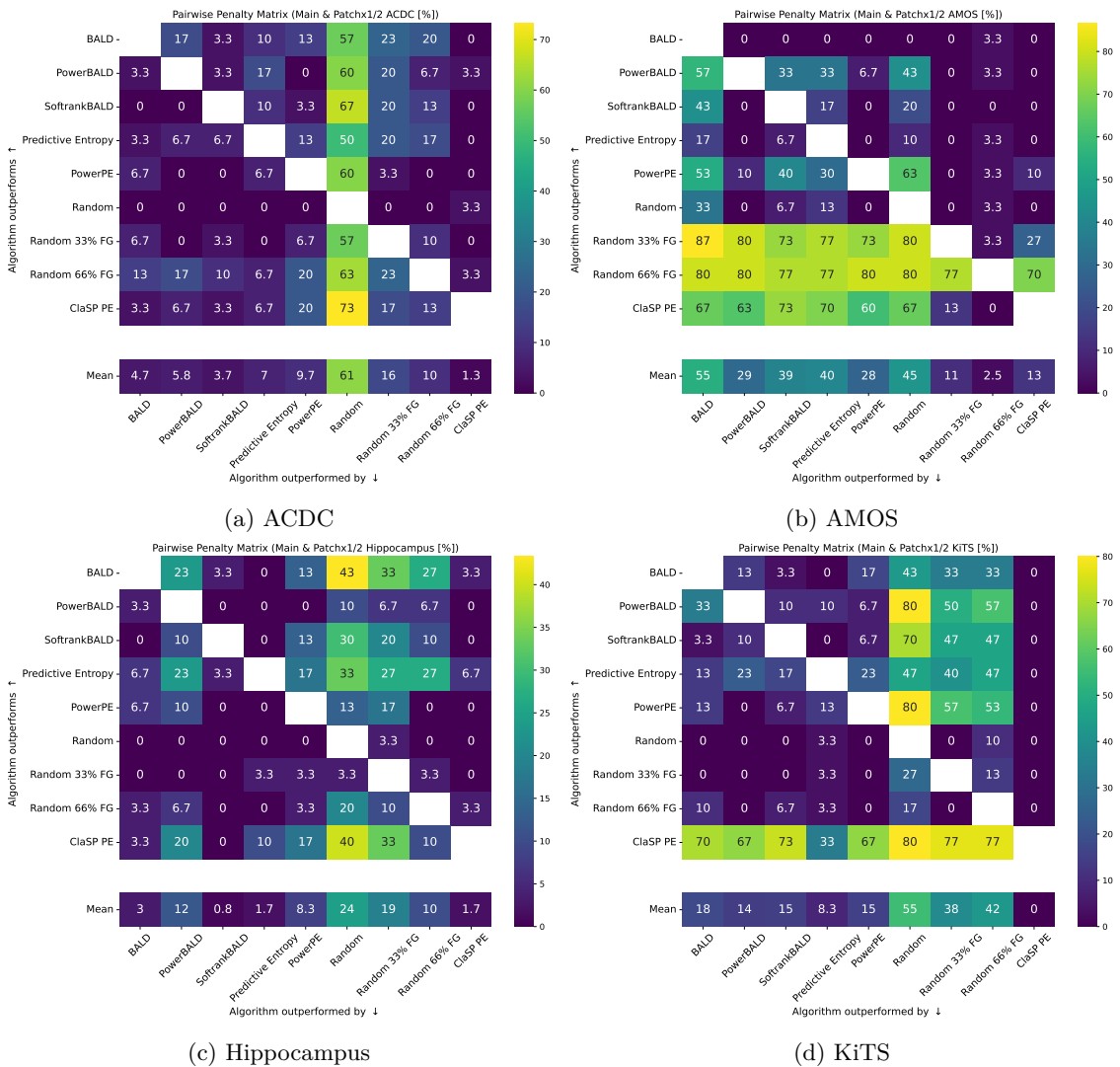

(a) ACDC

(b) AMOS

(c) Hippocampus

(d) KiTS

Figure 9: Pairwise Penalty Matrices aggregated over all Label Regimes and both query patch sizes for each dataset.

## E.2   Main Benchmark Results

The results shown in this section are obtained on the nnActive main study settings. Detailed results for AUBC, Final Dice, and FG-Eff, including standard deviations based on four seeds, are provided in table 6. The table includes results for the methods Cla PE 66% and 33%, as assessed in section 4.2. The overall PPM is shown in fig. 10, the respective dataset-specific PPMs are in fig. 11.

Table 6: Fine-grained Results for the nnActive Main Study for each dataset. Higher values are better, and colorization goes from dark green (best) to white (worst) with linear interpolation. AUBC and Final Dice are multiplied ×100 for improved readability. AUBC, Final, and FG-Eff can only be directly compared within each Label Regime on each dataset. The respective dataset characteristics are detailed in table 3.

(a) ACDC

| Dataset
Label Regime
Metric
Query Method | ACDC | | | | | | | | |
| --- | --- | --- | --- | --- | --- | --- | --- | --- | --- |
| | Low | | | Medium | | | High | | |
| | AUBC | Final Dice | FG-Eff | AUBC | Final Dice | FG-Eff | AUBC | Final Dice | FG-Eff |
| BALD | 79.84 ± 0.59 | 86.44 ± 0.96 | 26.98 ± 3.11 | 85.85 ± 0.45 | 89.62 ± 0.15 | 21.91 ± 4.20 | 87.74 ± 0.38 | 90.47 ± 0.18 | 15.09 ± 1.14 |
| PowerBALD | 81.18 ± 0.58 | 86.46 ± 0.55 | 46.29 ± 13.10 | 85.63 ± 0.37 | 89.07 ± 0.21 | 27.75 ± 4.00 | 87.50 ± 0.44 | 89.80 ± 0.17 | 17.94 ± 1.83 |
| SoftrankBALD | 80.71 ± 0.92 | 86.50 ± 0.95 | 35.71 ± 7.09 | 85.89 ± 0.49 | 89.33 ± 0.27 | 26.33 ± 5.01 | 87.28 ± 0.68 | 90.17 ± 0.14 | 14.53 ± 1.33 |
| Predictive Entropy | 80.02 ± 1.54 | 86.54 ± 0.95 | 26.49 ± 4.40 | 85.53 ± 0.59 | 89.42 ± 0.07 | 21.16 ± 3.11 | 87.65 ± 0.27 | 90.52 ± 0.06 | 13.58 ± 1.22 |
| PowerPE | 80.46 ± 0.30 | 86.56 ± 0.40 | 47.88 ± 14.09 | 85.24 ± 0.69 | 89.05 ± 0.22 | 27.92 ± 5.01 | 87.21 ± 0.60 | 89.67 ± 0.15 | 16.55 ± 1.18 |
| Random | 76.65 ± 0.81 | 80.34 ± 1.64 | 59.25 ± 33.53 | 82.24 ± 1.25 | 83.46 ± 0.87 | 38.22 ± 8.43 | 84.69 ± 0.96 | 86.28 ± 1.08 | 21.69 ± 3.79 |
| Random 33% FG | 81.28 ± 0.56 | 85.09 ± 1.14 | 40.88 ± 9.71 | 84.61 ± 0.65 | 87.51 ± 0.56 | 21.26 ± 1.49 | 86.95 ± 0.74 | 89.06 ± 0.44 | 15.81 ± 1.41 |
| Random 66% FG | 82.32 ± 0.33 | 86.70 ± 0.48 | 31.20 ± 4.32 | 86.16 ± 0.44 | 88.62 ± 0.52 | 18.95 ± 2.13 | 87.86 ± 0.33 | 89.94 ± 0.09 | 13.44 ± 0.79 |
| Cla PE 33% | 81.00 ± 0.74 | 86.38 ± 0.84 | 28.70 ± 2.71 | 85.67 ± 0.55 | 89.57 ± 0.09 | 19.93 ± 2.28 | 87.83 ± 0.37 | 90.50 ± 0.20 | 14.04 ± 0.92 |
| Cla PE 66% | 82.12 ± 0.71 | 87.45 ± 0.87 | 28.30 ± 2.47 | 86.11 ± 0.23 | 89.66 ± 0.15 | 18.04 ± 1.39 | 88.05 ± 0.15 | 90.55 ± 0.06 | 13.86 ± 1.00 |
| ClaP PE | 80.40 ± 0.55 | 86.11 ± 0.50 | 39.52 ± 8.07 | 86.33 ± 0.67 | 89.27 ± 0.47 | 33.61 ± 6.39 | 87.67 ± 0.35 | 89.97 ± 0.12 | 19.77 ± 2.01 |
| ClaSP PE | 81.31 ± 0.47 | 86.88 ± 0.78 | 37.36 ± 7.41 | 86.44 ± 0.67 | 89.50 ± 0.31 | 31.97 ± 11.86 | 87.91 ± 0.36 | 90.56 ± 0.09 | 18.66 ± 3.43 |

(b) AMOS

| Dataset
Label Regime
Metric
Query Method | AMOS | | | | | | | | |
| --- | --- | --- | --- | --- | --- | --- | --- | --- | --- |
| | Low | | | Medium | | | High | | |
| | AUBC | Final Dice | FG-Eff | AUBC | Final Dice | FG-Eff | AUBC | Final Dice | FG-Eff |
| BALD | 38.69 ± 2.34 | 34.05 ± 1.58 | -22.65 ± 8.50 | 52.56 ± 2.74 | 59.26 ± 2.73 | 1.49 ± 0.22 | 69.38 ± 0.70 | 74.95 ± 2.38 | -0.45 ± 0.20 |
| PowerBALD | 50.34 ± 3.00 | 56.18 ± 1.24 | 3.67 ± 14.54 | 66.11 ± 1.47 | 73.02 ± 2.01 | 18.19 ± 0.44 | 77.86 ± 0.14 | 80.48 ± 0.48 | 8.78 ± 0.08 |
| SoftrankBALD | 44.49 ± 1.56 | 45.75 ± 0.95 | -11.37 ± 4.19 | 60.01 ± 0.69 | 66.72 ± 0.65 | 5.66 ± 0.10 | 75.29 ± 1.46 | 81.23 ± 1.18 | 3.51 ± 0.39 |
| Predictive Entropy | 38.02 ± 3.35 | 39.19 ± 6.79 | -17.91 ± 8.48 | 56.30 ± 1.78 | 62.07 ± 1.39 | 2.62 ± 0.17 | 71.27 ± 1.52 | 80.79 ± 2.07 | 1.01 ± 0.41 |
| PowerPE | 47.66 ± 2.50 | 50.04 ± 2.30 | -9.78 ± 12.12 | 66.74 ± 2.80 | 73.68 ± 0.92 | 18.51 ± 1.17 | 77.92 ± 0.29 | 80.52 ± 0.16 | 8.86 ± 0.10 |
| Random | 42.26 ± 2.55 | 36.36 ± 2.92 | -134.74 ± 88.92 | 54.65 ± 2.82 | 56.22 ± 4.61 | 10.09 ± 3.26 | 73.82 ± 0.50 | 75.48 ± 0.37 | 7.33 ± 0.62 |
| Random 33% FG | 58.05 ± 1.54 | 62.95 ± 1.03 | 35.47 ± 11.41 | 71.78 ± 1.16 | 78.60 ± 0.37 | 36.44 ± 2.94 | 79.53 ± 0.38 | 82.68 ± 0.19 | 14.42 ± 0.47 |
| Random 66% FG | 62.84 ± 1.88 | 71.11 ± 1.42 | 43.64 ± 9.81 | 74.87 ± 0.64 | 80.72 ± 0.54 | 32.50 ± 6.08 | 80.98 ± 0.19 | 83.81 ± 0.32 | 12.32 ± 0.43 |
| Cla PE 33% | 45.98 ± 2.14 | 49.85 ± 1.01 | -6.04 ± 3.50 | 64.20 ± 2.09 | 71.54 ± 3.62 | 6.62 ± 0.21 | 79.52 ± 0.49 | 83.57 ± 0.39 | 5.96 ± 0.04 |
| Cla PE 66% | 51.66 ± 1.49 | 53.35 ± 1.75 | 1.00 ± 1.10 | 68.90 ± 1.71 | 78.50 ± 0.92 | 10.22 ± 0.26 | 80.84 ± 0.18 | 84.70 ± 0.07 | 7.47 ± 0.04 |
| ClaP PE | 53.60 ± 2.03 | 59.60 ± 3.92 | 15.86 ± 13.73 | 70.61 ± 1.45 | 78.51 ± 0.52 | 25.17 ± 0.97 | 79.83 ± 0.24 | 83.22 ± 0.26 | 11.34 ± 0.27 |
| ClaSP PE | 54.15 ± 2.26 | 59.82 ± 4.15 | 11.56 ± 6.25 | 71.28 ± 1.23 | 79.54 ± 0.29 | 20.01 ± 2.24 | 80.63 ± 0.12 | 84.40 ± 0.18 | 10.62 ± 0.60 |

(c) Hippocampus

| Dataset
Label Regime
Metric
Query Method | Hippocampus | | | | | | | | |
| --- | --- | --- | --- | --- | --- | --- | --- | --- | --- |
| | Low | | | Medium | | | High | | |
| | AUBC | Final Dice | FG-Eff | AUBC | Final Dice | FG-Eff | AUBC | Final Dice | FG-Eff |
| BALD | 88.46 ± 0.03 | 88.87 ± 0.06 | 9.58 ± 0.98 | 88.79 ± 0.02 | 89.18 ± 0.07 | 4.52 ± 0.06 | 89.03 ± 0.05 | 89.42 ± 0.05 | 3.49 ± 0.12 |
| PowerBALD | 88.20 ± 0.08 | 88.77 ± 0.11 | 9.21 ± 0.49 | 88.76 ± 0.04 | 89.16 ± 0.06 | 5.56 ± 0.07 | 88.98 ± 0.07 | 89.29 ± 0.10 | 3.90 ± 0.15 |
| SoftrankBALD | 88.44 ± 0.11 | 88.93 ± 0.18 | 9.61 ± 0.98 | 88.72 ± 0.08 | 89.12 ± 0.02 | 3.90 ± 0.05 | 89.03 ± 0.06 | 89.42 ± 0.07 | 3.60 ± 0.12 |
| Predictive Entropy | 88.50 ± 0.06 | 88.90 ± 0.10 | 9.75 ± 1.01 | 88.81 ± 0.04 | 89.18 ± 0.07 | 4.23 ± 0.06 | 89.07 ± 0.07 | 89.54 ± 0.03 | 3.73 ± 0.19 |
| PowerPE | 88.16 ± 0.08 | 88.70 ± 0.11 | 9.25 ± 0.52 | 88.63 ± 0.09 | 89.07 ± 0.21 | 4.41 ± 0.10 | 88.97 ± 0.07 | 89.33 ± 0.18 | 4.08 ± 0.24 |
| Random | 88.07 ± 0.10 | 88.58 ± 0.08 | 8.76 ± 0.47 | 88.65 ± 0.11 | 89.07 ± 0.04 | 5.10 ± 0.08 | 88.96 ± 0.09 | 89.29 ± 0.20 | 4.41 ± 0.25 |
| Random 33% FG | 88.22 ± 0.16 | 88.70 ± 0.08 | 9.60 ± 0.81 | 88.77 ± 0.13 | 89.22 ± 0.14 | 6.21 ± 0.17 | 88.94 ± 0.06 | 89.33 ± 0.10 | 3.85 ± 0.15 |
| Random 66% FG | 88.28 ± 0.13 | 88.76 ± 0.14 | 9.88 ± 0.73 | 88.63 ± 0.02 | 89.02 ± 0.04 | 4.21 ± 0.03 | 88.92 ± 0.08 | 89.26 ± 0.06 | 3.33 ± 0.11 |
| Cla PE 33% | 88.49 ± 0.06 | 88.97 ± 0.20 | 9.73 ± 0.94 | 88.88 ± 0.05 | 89.22 ± 0.08 | 5.21 ± 0.08 | 89.04 ± 0.05 | 89.43 ± 0.00 | 3.48 ± 0.21 |
| Cla PE 66% | 88.43 ± 0.10 | 88.90 ± 0.14 | 8.99 ± 0.64 | 88.77 ± 0.03 | 89.08 ± 0.12 | 4.02 ± 0.08 | 89.03 ± 0.06 | 89.46 ± 0.08 | 3.51 ± 0.14 |
| ClaP PE | 88.21 ± 0.13 | 88.64 ± 0.14 | 9.27 ± 0.71 | 88.69 ± 0.06 | 89.11 ± 0.08 | 5.28 ± 0.08 | 88.91 ± 0.07 | 89.25 ± 0.06 | 3.36 ± 0.11 |
| ClaSP PE | 88.28 ± 0.12 | 88.89 ± 0.13 | 9.59 ± 0.71 | 88.70 ± 0.11 | 89.15 ± 0.14 | 4.79 ± 0.11 | 88.97 ± 0.11 | 89.41 ± 0.09 | 3.86 ± 0.22 |

(d) KiTS

| Dataset
Label Regime
Metric
Query Method | KiTS | | | | | | | | |
| --- | --- | --- | --- | --- | --- | --- | --- | --- | --- |
| | Low | | | Medium | | | High | | |
| | AUBC | Final Dice | FG-Eff | AUBC | Final Dice | FG-Eff | AUBC | Final Dice | FG-Eff |
| BALD | 40.58 ± 2.75 | 44.03 ± 3.18 | 7.96 ± 0.82 | 55.06 ± 1.20 | 61.97 ± 1.49 | 6.51 ± 0.14 | 62.53 ± 0.84 | 67.57 ± 1.72 | 9.37 ± 0.46 |
| PowerBALD | 45.10 ± 2.91 | 47.67 ± 3.63 | 25.24 ± 6.06 | 54.53 ± 1.40 | 59.51 ± 1.15 | 10.16 ± 0.41 | 61.24 ± 0.57 | 65.04 ± 0.81 | 11.92 ± 0.64 |
| SoftrankBALD | 42.87 ± 2.91 | 47.12 ± 3.34 | 12.41 ± 2.03 | 54.83 ± 1.79 | 61.44 ± 2.02 | 6.99 ± 0.27 | 62.49 ± 0.74 | 67.00 ± 0.97 | 9.84 ± 0.66 |
| Predictive Entropy | 40.62 ± 2.74 | 45.53 ± 3.57 | 7.05 ± 0.64 | 57.42 ± 0.54 | 65.39 ± 0.51 | 6.19 ± 0.10 | 64.00 ± 0.15 | 68.74 ± 0.65 | 7.84 ± 0.21 |
| PowerPE | 45.30 ± 2.05 | 49.62 ± 1.13 | 28.70 ± 3.74 | 54.76 ± 1.10 | 58.67 ± 1.53 | 9.68 ± 0.28 | 60.66 ± 0.66 | 63.62 ± 1.19 | 9.62 ± 0.51 |
| Random | 38.75 ± 3.36 | 39.19 ± 4.13 | 28.47 ± 19.48 | 47.82 ± 1.84 | 48.41 ± 1.99 | 4.03 ± 2.75 | 53.80 ± 0.68 | 55.12 ± 1.27 | 8.93 ± 1.22 |
| Random 33% FG | 43.70 ± 0.87 | 47.35 ± 2.10 | 16.19 ± 1.33 | 51.50 ± 1.97 | 54.08 ± 2.76 | 3.27 ± 0.15 | 55.30 ± 1.26 | 56.79 ± 1.02 | 1.88 ± 0.04 |
| Random 66% FG | 44.97 ± 2.01 | 46.83 ± 2.53 | 11.28 ± 1.30 | 50.78 ± 0.97 | 51.67 ± 2.31 | 1.24 ± 0.02 | 53.73 ± 1.78 | 55.90 ± 0.84 | 0.68 ± 0.01 |
| Cla PE 33% | 45.62 ± 2.32 | 53.07 ± 1.36 | 12.70 ± 0.60 | 59.63 ± 0.73 | 66.41 ± 0.98 | 7.51 ± 0.07 | 64.82 ± 0.42 | 69.09 ± 0.49 | 8.73 ± 0.30 |
| Cla PE 66% | 48.09 ± 2.00 | 54.30 ± 2.46 | 13.97 ± 0.65 | 61.27 ± 0.63 | 68.42 ± 0.46 | 8.08 ± 0.09 | 65.58 ± 0.62 | 69.60 ± 0.35 | 8.70 ± 0.23 |
| ClaP PE | 46.80 ± 1.96 | 52.72 ± 1.65 | 29.08 ± 3.47 | 59.22 ± 1.46 | 63.91 ± 1.21 | 12.82 ± 0.75 | 63.74 ± 0.28 | 67.68 ± 0.85 | 11.66 ± 0.71 |
| ClaSP PE | 47.77 ± 1.63 | 54.83 ± 1.70 | 18.49 ± 2.11 | 60.33 ± 0.87 | 66.97 ± 0.91 | 10.38 ± 0.70 | 64.50 ± 0.29 | 69.53 ± 0.68 | 11.20 ± 1.25 |

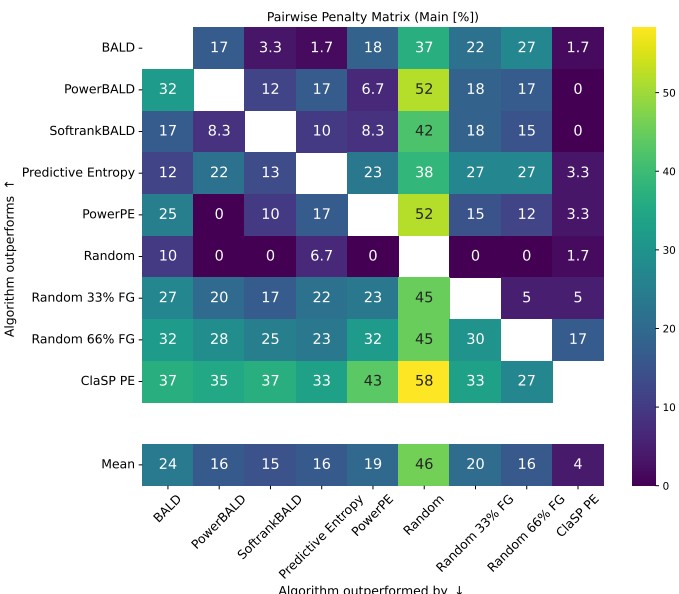

Figure 10: PPM aggregated over the nnActive main study experiments.

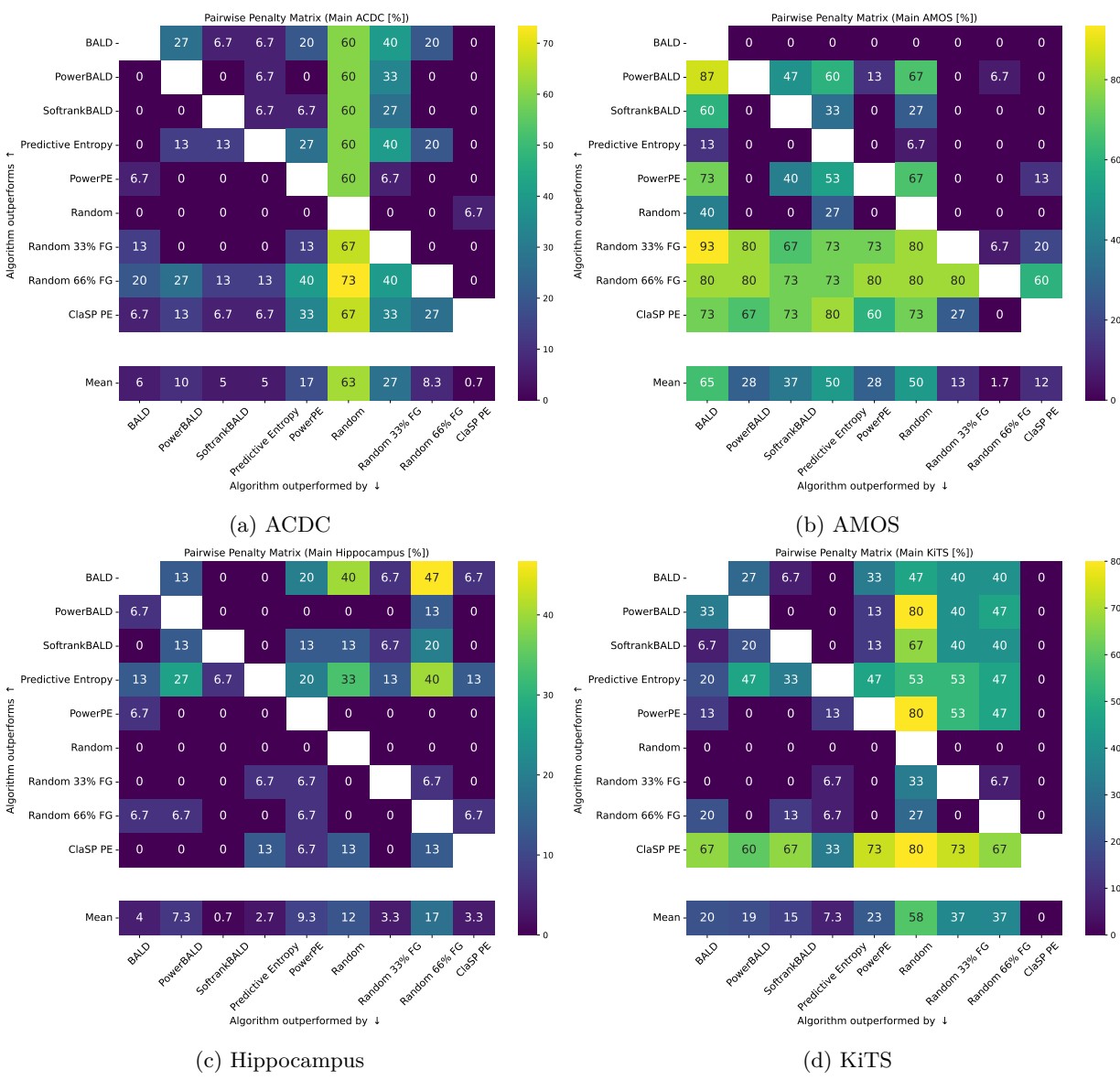

Figure 11: Pairwise Penalty Matrix aggregated over all Label Regimes for each dataset of the main study.

## E.3 Patch$\times\frac{1}{2}$ Setting results

Analogous to appendix E.2, this section provides results for the Patch$\times\frac{1}{2}$ settings, including a detailed results table (table 7), and the overall (fig. 12) and datset-specific (fig. 13) PPMs. The specific query patch sizes as well as further dataset characteristics are detailed in table 3.

Table 7: Fine-grained Results for the patch ablation with setting Patch$\times\frac{1}{2}$ for each dataset. Higher values are better, and colorization goes from dark green (best) to white (worst) with linear interpolation. AUBC and Final Dice are multiplied $\times100$ for improved readability. AUBC, Final, and FG-Eff can only be directly compared for each Label Regime on each dataset. The respective dataset characteristics are detailed in table 3.

(a) ACDC

| Dataset | ACDC | | | | | | | | |
| Label Regime | Low | | | Medium | | | High | | |
| Metric | AUBC | Final Dice | FG-Eff | AUBC | Final Dice | FG-Eff | AUBC | Final Dice | FG-Eff |
| Query Method | | | | | | | | | |
|---|---|---|---|---|---|---|---|---|---|
| BALD | 68.23 ± 2.31 | 77.72 ± 1.46 | 230.38 ± 440.89 | 75.80 ± 1.10 | 82.59 ± 1.24 | 149.81 ± 438.18 | 79.59 ± 1.05 | 83.99 ± 0.85 | 75.49 ± 69.33 |
| PowerBALD | 65.90 ± 5.61 | 75.24 ± 2.32 | 305.90 ± 942.82 | 77.07 ± 1.11 | 83.01 ± 1.21 | 207.99 ± 497.01 | 80.27 ± 1.39 | 84.54 ± 1.25 | 121.51 ± 193.59 |
| SoftrankBALD | 66.81 ± 3.68 | 75.98 ± 0.13 | 245.20 ± 442.21 | 76.84 ± 1.31 | 82.16 ± 0.47 | 199.23 ± 461.60 | 79.69 ± 1.03 | 84.03 ± 1.56 | 118.32 ± 181.98 |
| Predictive Entropy | 65.27 ± 2.45 | 75.79 ± 2.45 | 184.91 ± 201.36 | 74.67 ± 1.26 | 81.18 ± 1.32 | 119.52 ± 162.65 | 79.25 ± 0.95 | 83.58 ± 1.33 | 84.96 ± 102.80 |
| PowerPE | 65.70 ± 3.90 | 74.46 ± 2.28 | 300.97 ± 888.93 | 76.26 ± 2.36 | 82.16 ± 2.15 | 211.91 ± 452.95 | 79.85 ± 1.31 | 84.48 ± 1.55 | 132.75 ± 269.80 |
| Random | 59.38 ± 5.56 | 65.19 ± 4.17 | 479.15 ± 2311.84 | 70.99 ± 3.17 | 76.66 ± 1.22 | 461.53 ± 714.23 | 76.30 ± 0.80 | 79.09 ± 0.46 | 260.93 ± 529.78 |
| Random 33% FG | 67.98 ± 1.51 | 77.43 ± 0.27 | 216.14 ± 96.27 | 75.09 ± 1.78 | 81.65 ± 1.01 | 127.87 ± 65.39 | 79.55 ± 1.12 | 84.44 ± 0.32 | 87.88 ± 27.50 |
| Random 66% FG | 64.33 ± 1.17 | 73.97 ± 0.55 | 101.64 ± 46.52 | 74.69 ± 0.28 | 82.18 ± 1.52 | 72.09 ± 21.05 | 80.33 ± 0.56 | 85.88 ± 0.64 | 56.90 ± 8.42 |
| ClaSP PE | 68.00 ± 4.45 | 76.43 ± 2.90 | 239.84 ± 935.98 | 75.99 ± 2.64 | 82.60 ± 1.68 | 143.57 ± 329.15 | 79.82 ± 1.09 | 84.24 ± 0.79 | 88.31 ± 136.97 |

(b) AMOS

| Dataset | AMOS | | | | | | | | |
| Label Regime | Low | | | Medium | | | High | | |
| Metric | AUBC | Final Dice | FG-Eff | AUBC | Final Dice | FG-Eff | AUBC | Final Dice | FG-Eff |
| Query Method | | | | | | | | | |
|---|---|---|---|---|---|---|---|---|---|
| BALD | 13.98 ± 1.24 | 10.96 ± 2.19 | -150.74 ± 374.12 | 16.93 ± 2.33 | 17.85 ± 4.60 | -10.09 ± 20.38 | 30.15 ± 1.72 | 27.72 ± 0.83 | -19.23 ± 3.35 |
| PowerBALD | 14.54 ± 2.70 | 11.74 ± 2.59 | -248.75 ± 772.72 | 21.71 ± 1.48 | 25.83 ± 2.25 | 9.30 ± 15.98 | 40.14 ± 1.86 | 42.40 ± 1.47 | 8.53 ± 5.05 |
| SoftrankBALD | 13.63 ± 2.69 | 11.39 ± 1.68 | -127.75 ± 359.40 | 19.95 ± 1.55 | 23.48 ± 3.19 | -0.60 ± 8.12 | 35.13 ± 2.40 | 39.37 ± 1.87 | -3.05 ± 5.38 |
| Predictive Entropy | 13.83 ± 2.12 | 12.28 ± 1.98 | -83.91 ± 260.52 | 24.61 ± 2.34 | 27.37 ± 5.21 | 7.21 ± 2.16 | 36.63 ± 6.19 | 43.86 ± 6.88 | 1.71 ± 4.83 |
| PowerPE | 15.18 ± 2.85 | 13.00 ± 4.65 | -212.38 ± 565.02 | 23.28 ± 1.26 | 27.05 ± 2.03 | 15.28 ± 8.68 | 43.20 ± 1.78 | 47.34 ± 3.06 | 18.72 ± 3.42 |
| Random | 12.78 ± 2.02 | 8.89 ± 1.91 | -942.96 ± 5829.09 | 16.14 ± 1.62 | 16.99 ± 3.32 | -89.28 ± 165.24 | 37.56 ± 1.57 | 37.28 ± 3.44 | -3.38 ± 23.05 |
| Random 33% FG | 22.10 ± 1.18 | 24.14 ± 4.37 | 14.60 ± 105.81 | 39.32 ± 2.68 | 51.61 ± 4.21 | 103.81 ± 24.46 | 56.68 ± 1.86 | 65.54 ± 1.82 | 63.64 ± 11.01 |
| Random 66% FG | 31.10 ± 2.19 | 39.70 ± 0.34 | 134.71 ± 67.16 | 48.12 ± 0.68 | 60.25 ± 0.45 | 94.92 ± 28.62 | 62.07 ± 0.73 | 70.71 ± 0.60 | 51.62 ± 8.54 |
| ClaSP PE | 20.32 ± 2.58 | 25.84 ± 2.35 | 6.94 ± 99.73 | 31.54 ± 2.77 | 43.85 ± 2.68 | 36.96 ± 8.94 | 53.15 ± 3.14 | 67.43 ± 1.35 | 32.86 ± 3.25 |

(c) Hippocampus

| Dataset | Hippocampus | | | | | | | | |
| Label Regime | Low | | | Medium | | | High | | |
| Metric | AUBC | Final Dice | FG-Eff | AUBC | Final Dice | FG-Eff | AUBC | Final Dice | FG-Eff |
| Query Method | | | | | | | | | |
|---|---|---|---|---|---|---|---|---|---|
| BALD | 86.42 ± 0.47 | 87.85 ± 0.15 | 72.18 ± 173.53 | 87.64 ± 0.17 | 88.43 ± 0.19 | 15.02 ± 2.49 | 87.99 ± 0.16 | 88.76 ± 0.08 | 12.46 ± 1.33 |
| PowerBALD | 86.07 ± 0.35 | 87.45 ± 0.37 | 79.12 ± 97.84 | 87.32 ± 0.04 | 88.12 ± 0.04 | 18.16 ± 1.28 | 87.82 ± 0.04 | 88.47 ± 0.07 | 17.28 ± 1.29 |
| SoftrankBALD | 86.44 ± 0.33 | 87.66 ± 0.32 | 73.07 ± 153.84 | 87.54 ± 0.17 | 88.27 ± 0.10 | 16.63 ± 2.19 | 87.92 ± 0.07 | 88.66 ± 0.14 | 15.13 ± 1.68 |
| Predictive Entropy | 86.34 ± 0.22 | 87.69 ± 0.09 | 63.64 ± 128.00 | 87.43 ± 0.14 | 88.41 ± 0.09 | 13.37 ± 1.22 | 87.99 ± 0.14 | 88.74 ± 0.09 | 12.30 ± 1.71 |
| PowerPE | 86.21 ± 0.70 | 87.56 ± 0.51 | 84.35 ± 144.15 | 87.43 ± 0.11 | 88.29 ± 0.11 | 19.81 ± 2.65 | 87.94 ± 0.11 | 88.43 ± 0.15 | 18.11 ± 2.41 |
| Random | 85.62 ± 0.65 | 86.74 ± 0.31 | 118.43 ± 222.57 | 87.06 ± 0.21 | 87.76 ± 0.10 | 25.64 ± 5.49 | 87.58 ± 0.15 | 88.13 ± 0.15 | 26.07 ± 3.15 |
| Random 33% FG | 85.69 ± 0.56 | 87.02 ± 0.19 | 78.79 ± 124.77 | 87.26 ± 0.17 | 88.00 ± 0.07 | 17.19 ± 2.51 | 87.74 ± 0.06 | 88.31 ± 0.11 | 15.41 ± 1.27 |
| Random 66% FG | 86.24 ± 0.13 | 87.54 ± 0.15 | 57.16 ± 55.78 | 87.49 ± 0.21 | 88.27 ± 0.10 | 14.91 ± 1.15 | 87.85 ± 0.21 | 88.54 ± 0.17 | 12.43 ± 0.64 |
| ClaSP PE | 86.62 ± 0.41 | 87.80 ± 0.28 | 79.67 ± 214.16 | 87.66 ± 0.06 | 88.43 ± 0.05 | 14.89 ± 2.08 | 87.96 ± 0.10 | 88.59 ± 0.08 | 12.64 ± 1.97 |

(d) KiTS

| Dataset | KiTS | | | | | | | | |
| Label Regime | Low | | | Medium | | | High | | |
| Metric | AUBC | Final Dice | FG-Eff | AUBC | Final Dice | FG-Eff | AUBC | Final Dice | FG-Eff |
| Query Method | | | | | | | | | |
|---|---|---|---|---|---|---|---|---|---|
| BALD | 25.10 ± 0.55 | 31.76 ± 4.51 | 86.79 ± 97.07 | 38.56 ± 3.27 | 43.25 ± 3.79 | 31.30 ± 10.14 | 48.46 ± 1.19 | 53.50 ± 1.28 | 23.96 ± 6.78 |
| PowerBALD | 27.91 ± 1.74 | 29.39 ± 1.30 | 184.72 ± 578.47 | 41.70 ± 1.09 | 45.59 ± 1.40 | 78.84 ± 44.77 | 49.60 ± 0.95 | 54.00 ± 1.25 | 43.02 ± 17.06 |
| SoftrankBALD | 25.67 ± 2.68 | 31.47 ± 1.54 | 90.70 ± 90.22 | 41.08 ± 1.00 | 46.14 ± 1.77 | 46.46 ± 16.84 | 49.08 ± 1.12 | 54.39 ± 1.53 | 31.73 ± 10.26 |
| Predictive Entropy | 24.08 ± 1.56 | 29.07 ± 5.82 | 41.77 ± 25.38 | 40.99 ± 3.00 | 46.80 ± 3.63 | 23.97 ± 2.94 | 50.22 ± 1.42 | 55.79 ± 1.07 | 14.86 ± 1.67 |
| PowerPE | 27.96 ± 3.53 | 30.88 ± 4.84 | 207.47 ± 651.68 | 42.26 ± 0.77 | 46.55 ± 0.95 | 83.36 ± 51.36 | 49.48 ± 1.57 | 53.59 ± 1.03 | 44.14 ± 21.87 |
| Random | 22.00 ± 1.62 | 22.85 ± 2.15 | 139.89 ± 529.13 | 35.14 ± 2.00 | 37.95 ± 1.74 | 93.75 ± 139.60 | 42.73 ± 1.09 | 44.35 ± 1.60 | 46.87 ± 73.85 |
| Random 33% FG | 23.88 ± 3.43 | 28.83 ± 1.46 | 49.00 ± 31.48 | 37.88 ± 0.74 | 41.24 ± 0.88 | 17.55 ± 1.47 | 42.28 ± 1.19 | 44.19 ± 1.31 | 3.48 ± 0.15 |
| Random 66% FG | 24.43 ± 1.96 | 28.80 ± 2.90 | 29.82 ± 7.76 | 34.12 ± 1.40 | 36.52 ± 1.24 | 4.34 ± 0.27 | 40.24 ± 1.31 | 42.58 ± 0.88 | 0.68 ± 0.04 |
| ClaSP PE | 32.16 ± 1.04 | 40.16 ± 0.63 | 98.74 ± 133.82 | 45.76 ± 0.53 | 52.66 ± 0.97 | 31.13 ± 15.63 | 53.69 ± 0.93 | 59.96 ± 2.14 | 19.75 ± 5.40 |

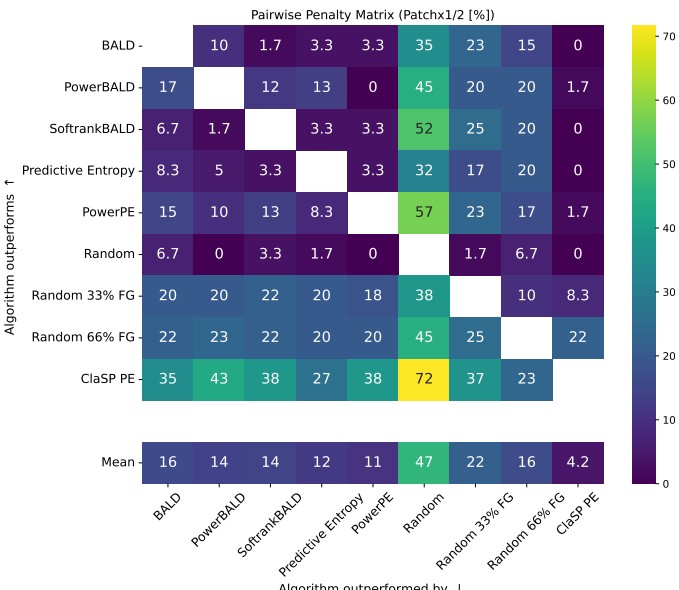

Figure 12: PPM aggregated over the Patch$\times\frac{1}{2}$ settings. Mean row results change compared to the nnActive main study (fig. 10).

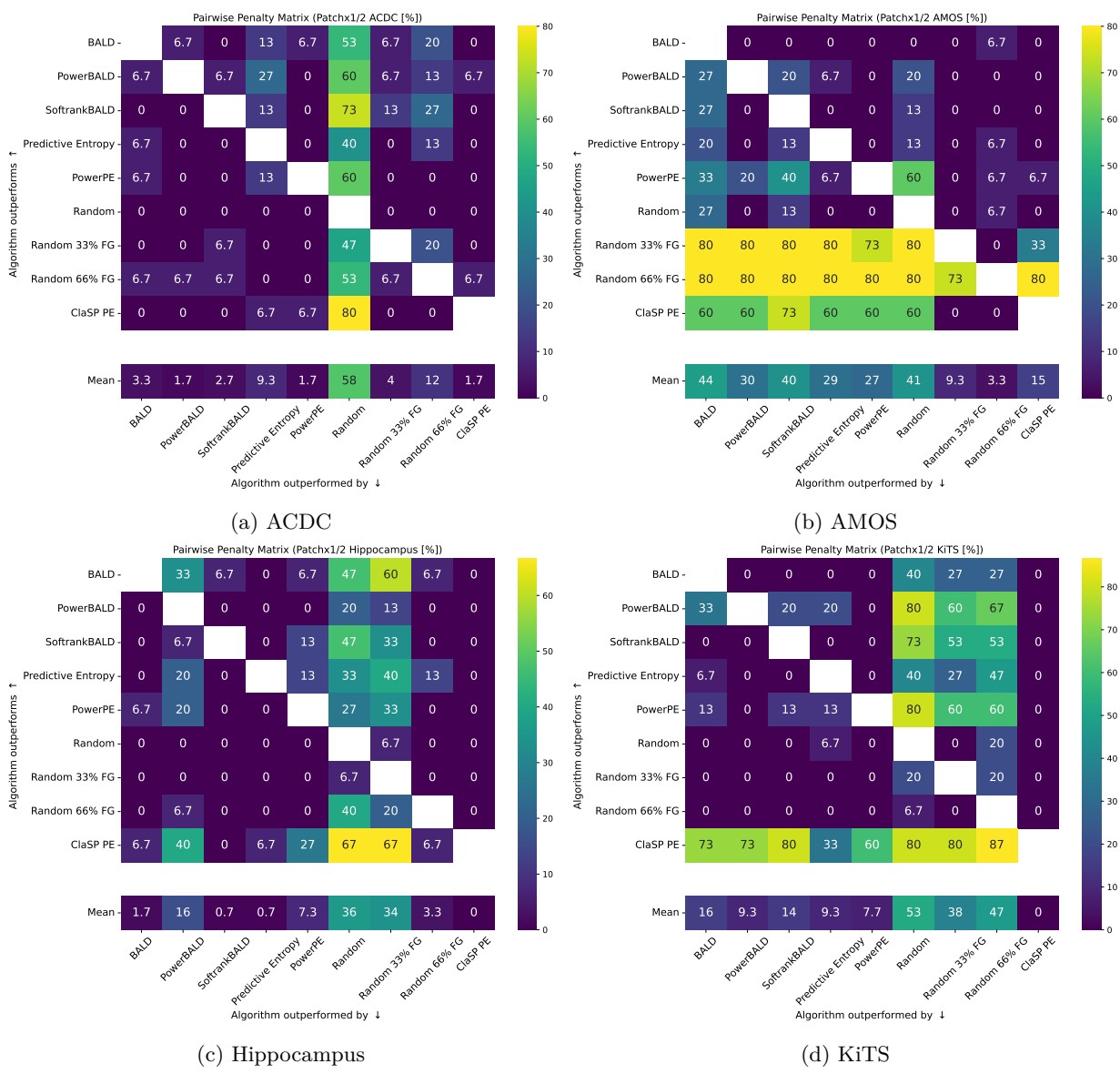

(a) ACDC

(b) AMOS

(c) Hippocampus

(d) KiTS

Figure 13: Pairwise Penalty Matrix aggregated over all Label Regimes for each dataset for the Patch$\times\frac{1}{2}$ ablation.

### E.4 500 Epochs Setting results

For the ablation on the loss scenarios on AMOS in section 4.1, we provide detailed results in table 8.

Table 8: Fine-grained Results for the AMOS experiments when training for 500 epochs. Higher values are better, and colorization goes from dark green (best) to white (worst) with linear interpolation. AUBC and Final Dice are multiplied ×100 for improved readability. AUBC, Final, and FG-Eff can only be directly compared for each Label Regime on each dataset.

| Dataset | | | | | AMOS | | | | |
|---|---|---|---|---|---|---|---|---|---|
| Label Regime | | Low | | | Medium | | | High | |
| Metric | AUBC | Final Dice | FG-Eff | AUBC | Final Dice | FG-Eff | AUBC | Final Dice | FG-Eff |
| Query Method | | | | | | | | | |
| BALD | – | – | – | 75.76 ± 1.20 | 81.67 ± 2.40 | 7.90 ± 0.54 | 84.37 ± 0.10 | 87.18 ± 0.07 | 7.71 ± 0.12 |
| PowerBALD | – | – | – | 79.87 ± 0.33 | 83.96 ± 0.41 | 24.98 ± 1.10 | 84.50 ± 0.16 | 86.35 ± 0.04 | 12.57 ± 0.26 |
| SoftrankBALD | – | – | – | 79.04 ± 0.29 | 83.55 ± 0.08 | 14.54 ± 0.33 | 84.60 ± 0.26 | 86.83 ± 0.16 | 9.27 ± 0.11 |
| Predictive Entropy | – | – | – | 77.21 ± 0.53 | 83.40 ± 0.41 | 9.19 ± 0.25 | 84.70 ± 0.03 | 87.52 ± 0.08 | 7.09 ± 0.09 |
| PowerPE | – | – | – | 79.27 ± 0.36 | 83.35 ± 0.21 | 22.80 ± 0.69 | 84.32 ± 0.32 | 86.23 ± 0.19 | 11.59 ± 0.27 |
| Random | – | – | – | – | – | – | – | – | – |
| Random 33% FG | 65.08 ± 1.59 | 71.22 ± 2.39 | 64.00 ± 23.13 | 79.41 ± 0.48 | 83.40 ± 0.35 | 33.90 ± 3.19 | 84.16 ± 0.14 | 86.04 ± 0.13 | 14.43 ± 0.49 |
| Random 66% FG | 68.76 ± 1.38 | 77.13 ± 0.55 | 60.89 ± 13.80 | 81.28 ± 0.51 | 85.11 ± 0.30 | 27.99 ± 2.92 | 85.10 ± 0.24 | 86.96 ± 0.24 | 12.61 ± 0.46 |
| Cla PE 66% | 65.18 ± 0.80 | 71.87 ± 0.85 | 24.29 ± 2.84 | 80.37 ± 0.56 | 85.23 ± 0.30 | 14.08 ± 0.21 | 85.38 ± 0.02 | 87.66 ± 0.11 | 8.69 ± 0.07 |
| ClaSP PE | 64.73 ± 0.30 | 73.44 ± 1.30 | 36.02 ± 6.94 | 80.74 ± 0.48 | 85.17 ± 0.24 | 23.55 ± 3.92 | 85.37 ± 0.15 | 87.63 ± 0.08 | 13.73 ± 0.96 |

### E.5 Analyzing ClaSP PE performance on AMOS on a class level

We analyze the performance of ClaSP PE relative to Random 66%FG on the Final Dice with regard to each class and the percentage of voxels queried on the AMOS dataset for 200 and 500 epochs on the main setting across all label regimes in fig. 14. We observe that the longer training leads to ClaSP PE gaining more performance than Random 66% FG, but it also leads to a general increase in queried foreground. Overall, the esophagus, postcava, pancreas, right adrenal gland, left adrenal gland and duodenum (classes 5, 9, 10, 11, 12 and 13) are the most challenging classes. Importantly, the performance differences for right and left adrenal gland reduce with larger annotation budgets and longer training. While the largest performance gains stem from the bladder and prostate (classes 14 and 15).

In the low-label regime, especially the classes 11 and 12 (right and left adrenal gland), lead to relative performance losses of ClaSP PE relative to Random 66% FG which are also less frequently queried. In the medium and low-label regime with more queries for these classes this performance difference diminishes for 200 epochs and vanishes for 500 epochs below random noise.

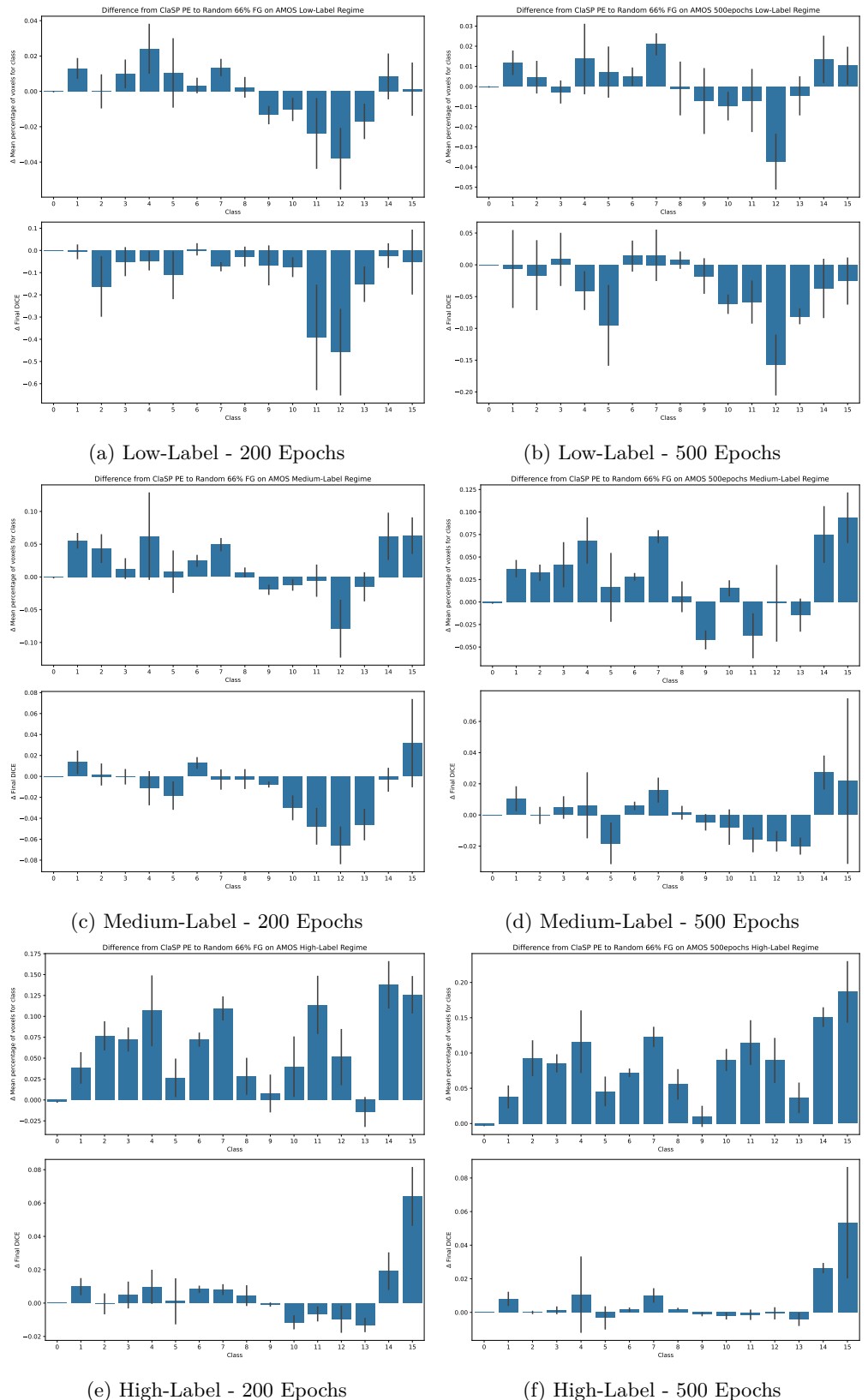

Figure 14: Visualization of the difference of the percentage of voxels for all classes alongside Final Dice performance on AMOS in the Main setting from ClaSP PE to Random 66% FG trained with 200 & 500 epochs. Error bars denote the Standard Deviation.

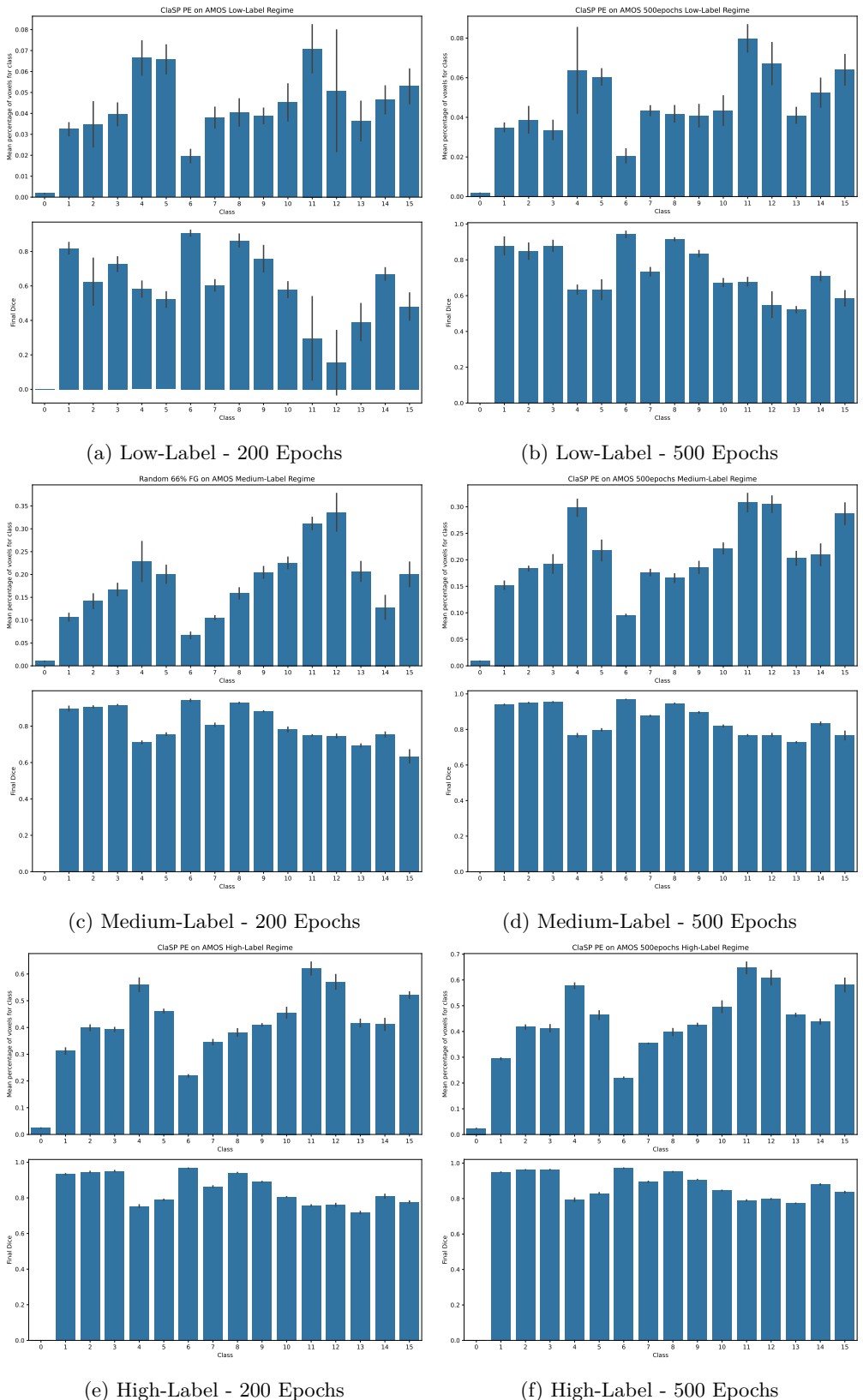

Figure 15: Visualization of the percentage of voxels for all classes alongside Final Dice performance on AMOS in the Main setting from ClaSP PE trained with 200 & 500 epochs. Error bars denote the Standard Deviation.

### E.6 Comparing Pairwise Penalty Matrix with different p-values

Here we will compare the results of the pairwise penalty matrix aggregated over the main and patch$\times\frac{1}{2}$ experiments when using p-values set to 0.05, 0.02 and corrected with the Holm-Bonferroni method (Holm, 1979) to reduce the probability of false rejections due to multiple tests for a p-value of 0.05.

The correction is performed for each singular experiment (one dataset and one label-regime) with the value $m$ used to divide the p-value:

$$m = \frac{\#\text{QMs} \times (\#\text{QMs} - 1)}{2} \times \#\text{Loops} = 180,$$

when using $\#\text{QMs} = 9$ and $\#\text{Loops} = 5$ based on our experiment setup. This computation of the $m$-value factors in each pairwise comparison.

In common practice, the pairwise penalty matrix is used without any corrections for multiple tests and with a fixed p-value of 0.05 (Ash et al., 2020; Föllmer et al., 2024; Lüth et al., 2023; 2025; Beck et al., 2021).

The results are shown in fig. 16. Overall, the trend for the mean row (lower is better) is similar for p-values 0.05 and 0.02, where ClaSP PE is the best performing method, followed by a group of QMs (Predictive Entropy, SoftrankBALD, PowerPE, and PowerBALD), Random 66% and 33% FG, with Random as the least performant method.

Meanwhile, for the Holm-Bonferroni method, ClaSP PE remains the best performing method, but now Predictive Entropy is the only other AL method that outperforms the random baselines.

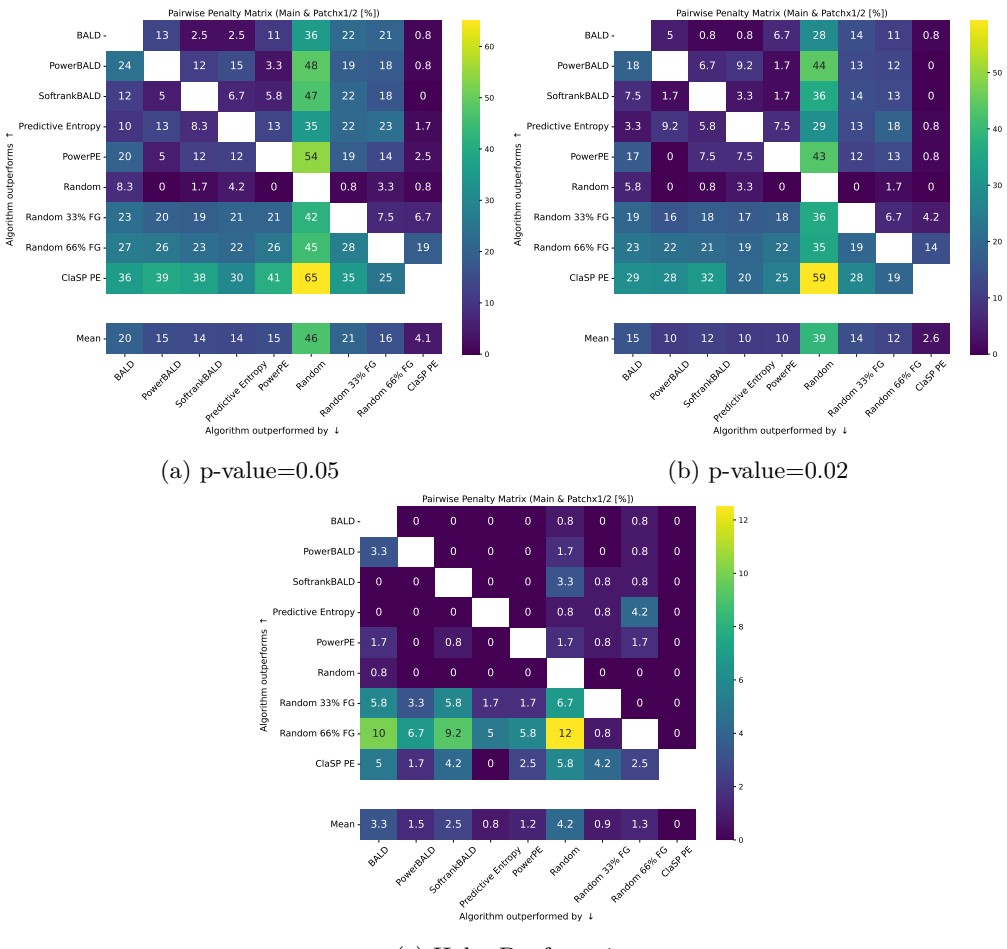

(a) p-value=0.05

(b) p-value=0.02

(c) Holm-Bonferroni

Figure 16: As expected, for smaller p-values, the number of significant results decreases, especially so for the Holm-Bonferroni method due to the number of comparisons (180). When discussing the overall trend based on the Mean row (lower is better), we observe that ClaSP PE exhibits the lowest fraction of scenarios where it is outperformed by another method. This trend is similar for p-values 0.05 and 0.02, where ClaSP PE is the best performing method, followed by a group of QMs (Predictive Entropy, SoftrankBALD, PowerPE, and PowerBALD), Random 66% and 33% FG, with Random as the least performant method. Meanwhile, for the Holm-Bonferroni method, ClaSP PE remains the best performing method, but now Predictive Entropy is the only other AL method that outperforms the random baselines.

### E.7  Mean Performance estimate

The computation for the values in table 1 are obtained for the mean $\overline{\mu}$:

$$\overline{\mu} = \frac{1}{S} \sum_{s=1}^{S} \mu_s,$$

where $\mu_s$ is the mean performance for each setting. The standard error is obtained using the gaussian error propagation for the mean errors:

$$\Delta\overline{\mu} = \frac{1}{S} \sqrt{\sum_{s=1}^{S} (\Delta\mu_s)^2},$$

where $\Delta\mu_s = \frac{1}{\sqrt{N}}\sigma_s$, $N$ represents the number of seeds per experiment and $\sigma_s$ is the standard deviation for a single experiment.

We use as 95% confidence interval $1.96 \times \Delta\overline{\mu}$.

# F    Guidelines for Real-World Deployment of ClaSP PE

In the following, we provide details on the systematic selection of query patch size and query budget parameters for applying ClaSP PE to unseen datasets. The parameters that we obtain for the roll-out datasets are provided in table 4. Despite our extensive validation, demonstrating the generalization capabilities, we can not guarantee good model performance beyond the tested settings, especially for lower fractions of annotated data.

**Query Size Selection**  We recommend to normalize the query size based on the number of foreground classes in the dataset to optimally leverage the performance gains through class-stratified sampling. In our roll-out experiments, we calculate the total query budget (per AL cycle) by counting contribution of 50 or 100 patches per class, depending on the task complexity of segmenting certain target structures. Specifically, we use a query budget contribution of 100 for classes where we expect higher variance, such as the tumor class compared to the liver class in LiTS.

**Number of AL Loops**  In our experiments we performed five AL loops. Since AL performance typically improves or remains stable relative to random strategies in later stages or with larger annotation budgets, extending the number of AL loops is generally safe and may further improve performance. We hypothesize that leaving the value for $\beta$ (inverse power noising strength) after the 5th loop at 100 should be sufficient for ClaSP PE, as the segmentation model should by then be able to effectively exploit its understanding of the task. Crucially, we do not advise to reduce the number of AL loops.

**Starting Budget Selection**  The starting budgets should be selected with a mix of completely random patches and a random class balanced selection of patches. We used a factor of 33% of patches which surely feature foreground. In practice, the patches which surely feature foreground can be simply obtained by going into random images and just selecting some patches featuring target structures of interest while counting the amount of patches to ensure that they are somewhat class balanced.

**Query Patch Size Selection**  For the size of the query patches, we recommend choosing the median size of the target structures, in order to obtain representative samples. This is realizable in practice, as an estimation of the object sizes can typically be done efficiently, without necessitating the availablity of fine-grained annotations. In the benchmarking scenario, we proceed as follows: First, we compute the median bounding box size per class based on the largest connected component per image. Then, we take the overall median bounding box size across classes. An exception is the LiTS dataset, where we only consider the tumor class (omitting the liver class), as the liver is significantly larger while the tumor structures are of particular interest.

## G Roll-Out Results

Detailed results for the roll-out study in section 5 are provided in table 9. The corresponding PPM is shown in fig. 6.

Table 9: Fine-grained Results for the Roll-Out Scenario. Higher values are better, and colorization goes from dark green (best) to white (worst) with linear interpolation. AUBC and Final Dice are reported with a factor (×100) for improved readability. AUBC, Final, and FG-Eff can only be directly compared for each Label Regime on each dataset. The respective dataset characteristics are detailed in table 4.

| Dataset
Label Regime
Metric
Query Method | LiTS
Roll-Out | | | WORD
Roll-Out | | | Tooth Fairy 2
Roll-Out | | | MAMA MIA
Roll-Out | | |
|---|---|---|---|---|---|---|---|---|---|---|---|---|
| | AUBC | Final Dice | FG-Eff | AUBC | Final Dice | FG-Eff | AUBC | Final Dice | FG-Eff | AUBC | Final Dice | FG-Eff |
| Random | $51.23 \pm 1.21$ | $52.38 \pm 2.21$ | $46.25 \pm 45.84$ | $77.35 \pm 1.04$ | $78.03 \pm 0.88$ | $3.66 \pm 0.25$ | $61.83 \pm 0.25$ | $64.32 \pm 0.38$ | $11.88 \pm 0.18$ | $55.23 \pm 2.06$ | $58.24 \pm 1.90$ | $39.13 \pm 209.13$ |
| Random 66% FG | $48.63 \pm 1.22$ | $50.05 \pm 1.32$ | $1.27 \pm 0.15$ | $78.19 \pm 0.34$ | $78.25 \pm 0.15$ | $1.34 \pm 0.02$ | $65.30 \pm 0.28$ | $68.61 \pm 0.15$ | $10.85 \pm 0.17$ | $44.38 \pm 3.68$ | $45.10 \pm 5.64$ | $-4.67 \pm 0.91$ |
| Predictive Entropy | $57.81 \pm 1.17$ | $65.38 \pm 2.76$ | $38.94 \pm 4.50$ | $78.43 \pm 0.07$ | $78.96 \pm 0.28$ | $0.91 \pm 0.00$ | $66.65 \pm 0.60$ | $71.97 \pm 0.08$ | $16.25 \pm 0.60$ | $59.07 \pm 4.15$ | $64.74 \pm 2.42$ | $9.43 \pm 2.08$ |
| ClaSP PE | $60.30 \pm 1.74$ | $65.80 \pm 1.47$ | $39.60 \pm 12.95$ | $78.27 \pm 0.41$ | $78.42 \pm 0.17$ | $1.33 \pm 0.02$ | $67.32 \pm 0.37$ | $71.49 \pm 0.17$ | $20.07 \pm 0.43$ | $63.85 \pm 1.58$ | $68.62 \pm 1.36$ | $57.36 \pm 407.92$ |

## H Limitations

**Benchmark overfitting.** ClaSP PE was developed on the nnActive benchmark and therefore carries the risk of over-optimization. However, the general strong performance on the Roll-Out Study against the Predictive Entropy and Random FG 66% FG, which were the other best performing methods on the nnActive benchmark, shows its generalization capabilities to novel scenarios. We also wish to highlight that virtually all AL methods face the danger of being overdesigned for a specific benchmark as they necessitate design decisions that need to be evaluated empirically (Shi et al., 2024a; Föllmer et al., 2024; Gaillochet et al., 2023b; Vepa et al., 2024). The combined evaluation on the nnActive Benchmark and Roll-Out study, which to our knowledge is the most comprehensive to date, mitigates the risk of benchmark-specific overfitting relative to earlier approaches (Lüth et al., 2025). Further, the performance of ClaSP PE on the nnActive benchmark suggests generalization capabilities beyond our conservative Guidelines for Real-World Deployment. We encourage future benchmarking efforts of AL methods for 3D biomedical segmentation, demonstrating their generalization capabilities on novel datasets separate from method development, thereby reducing potential conflicts of interest.

Our empirical findings are specific to the nnActive framework using nnU-Net as the segmentation network and 3D patches as queries. Results may differ with alternative architectures, regularization techniques, self-supervised or semi-supervised learning approaches, or different query designs (2D slices or whole 3D images).

**Dependency on model predictive capacity.** ClaSP PE relies on the model producing sufficiently accurate multi-class segmentations, since these predictions underpin the stratified query selection. In the low-label regime of the AMOS dataset (see section 4.1 Query Design), we observed that limited initial labels can result in inadequate segmentation quality, reducing the effectiveness of stratification. Our final recommended guidelines for using ClaSP PE mitigate this risk by using a query size based on the number of classes that is most likely to lead to a sufficient initial segmentation quality, as exemplified by the results in the Roll-Out study. Moreover, the use of pre-trained foundation models may further improve early-stage segmentation quality (Gupte et al., 2024).

**Economic trade-offs of AL.** We wish to emphasize that AL inherently represents a wager with the aim of reducing the overall cost of building an ML pipeline where compute cost is better against an expected reduction in annotation effort (Settles, 2011). The decision whether to employ AL or not is an economic question dependent on multiple factors, such as annotation cost and computational resources. In this work, we demonstrate that ClaSP PE within the nnActive Framework shows strong evidence to reduce the annotation cost when employing AL in a wide variety of settings. However, the cost of employing AL needs to be evaluated in comparison to the expected gains, which is outside the scope of this work and represents a fruitful direction for future research.

**Comparison to more complex AL baselines.** While more sophisticated AL strategies (s.a. Hübotter et al. (2024); Föllmer et al. (2024)) could in principle yield similar gains, there is currently little evidence that they can be made practical in our setting. In particular, extending such methods to 3D biomedical image segmentation remains an unsolved challenge: querying 3D patches instead of 2D slices while integrating the full complexity of state-of-the-art segmentation pipelines poses substantial computational and algorithmic hurdles (see section 2). Our focus here was therefore on designing an approach that is robust and easily deployable across new datasets, leaving the open problem of adapting more advanced AL techniques to the 3D domain for future work.

**Hyperparameters of ClaSP PE.** We addressed the need for a class-balanced dataset and hard-to-predict cases, and early-stage diversification and later exploitation through careful balancing of the stratified sampling ratio $\alpha$ and a pre-defined scheduling for power-noising of $\beta$. We empirically rigorously validate these modifications to ensure overall benefits on a wide variety of datasets based on a large-scale benchmark for development and an evaluation on held-out roll-out datasets. However, our results also show that this exact setup is not always the optimal solution, and it might be even more favorable to have heuristics to adapt both $\alpha$ and the scheduling of $\beta$ based on dataset characteristics and other confounding information, such as model performance.

**Metric Dependence and Generalization** Our evaluation metrics are based on the mean Dice as a measure of segmentation performance, which assigns equal importance to all classes. We do not demonstrate that our method generalizes to scenarios where evaluation metrics weigh classes differently. Although mean Dice is widely used in medical image segmentation, there are tasks where not all classes are equally important. For example, class importance may be proportional to the frequency (micro-averaged performance) or dictated by clinical relevance (weighted macro-average). While the class-balanced sampling in ClaSP PE is naturally aligned with mean Dice, alternative sampling weights may be desirable when evaluation metrics assign unequal importance to classes. Finally, we emphasize that class-balanced sampling serves a broader purpose in maintaining semantic diversity across AL cycles, which remains beneficial independent of the chosen metric.

# I Qualitative Results

In this section, we provide additional qualitative results to demonstrate the effects of the class stratification used in ClaSP PE, as compared to standard Predictive Entropy.

## I.1 Query Visualization

In figs. 17 to 20, we provide exemplary visualizations of the queried patches of PE and ClaSP PE after the first AL loop on all nnActive benchmark datasets on the low-label regime using the main settings. For ACDC (fig. 17), the stratification of ClaSP PE leads to a more diverse query selection and more foreground being queried. Further, ClaSP PE mitigates the risk of an overly focus on prominent classes, such as the posterior hippocampus (fig. 18), the tumor class on KiTS (fig. 19), or the liver class on AMOS (fig. 20).

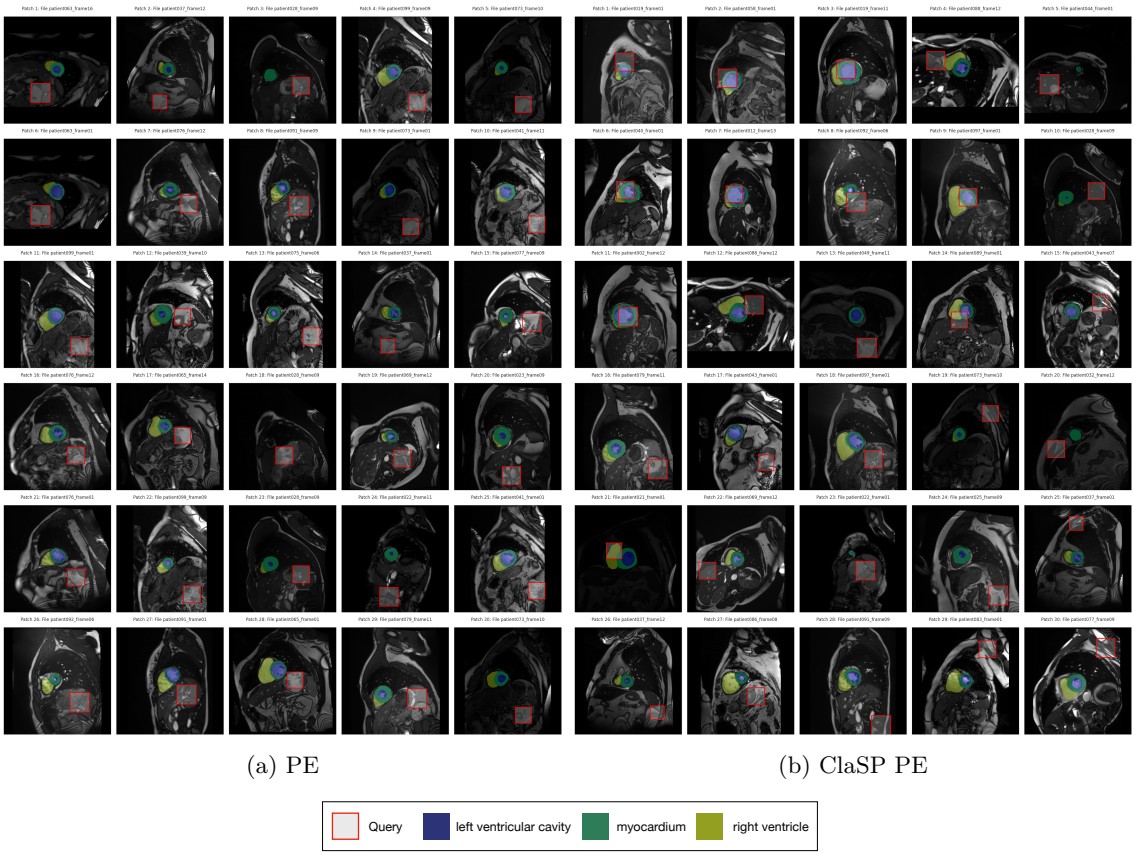

(a) PE           (b) ClaSP PE

Figure 17: Exemplary visualization of the queried patches using PE (a) and ClaSP PE (b) after the first AL loop on the ACDC dataset (same seed to ensure comparability). For 2D visualization, we selected the center slice of the 3D patches. Best viewed zoomed in.

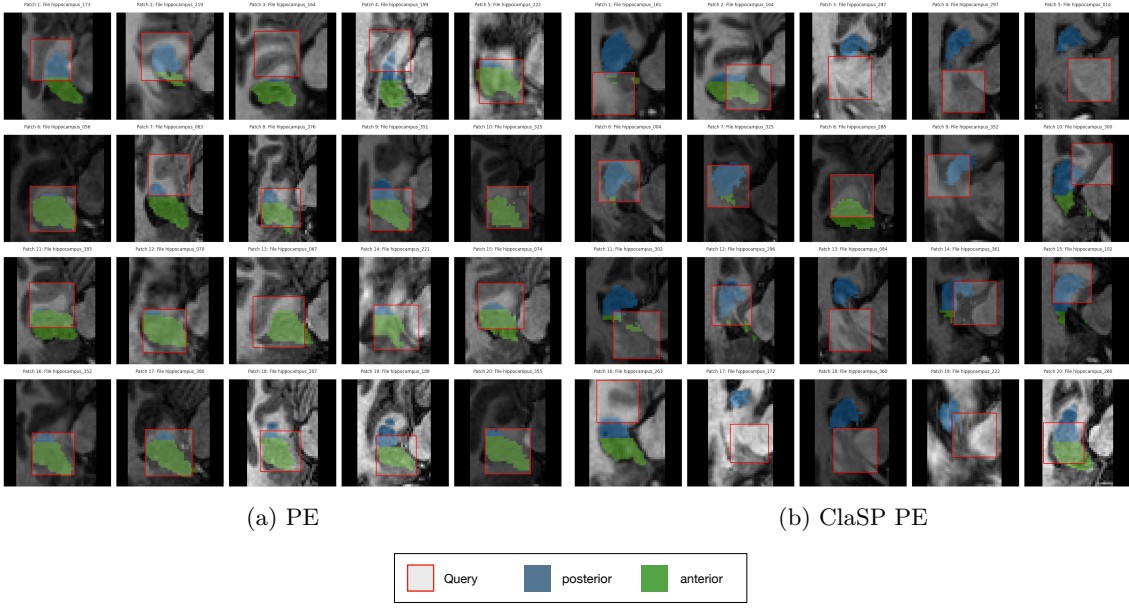

(a) PE  (b) ClaSP PE

Query  posterior  anterior

Figure 18: Exemplary visualization of the queried patches using PE (a) and ClaSP PE (b) after the first AL loop on the Hippocampus dataset (same seed to ensure comparability). For 2D visualization, we selected the center slice of the 3D patches. Best viewed zoomed in.

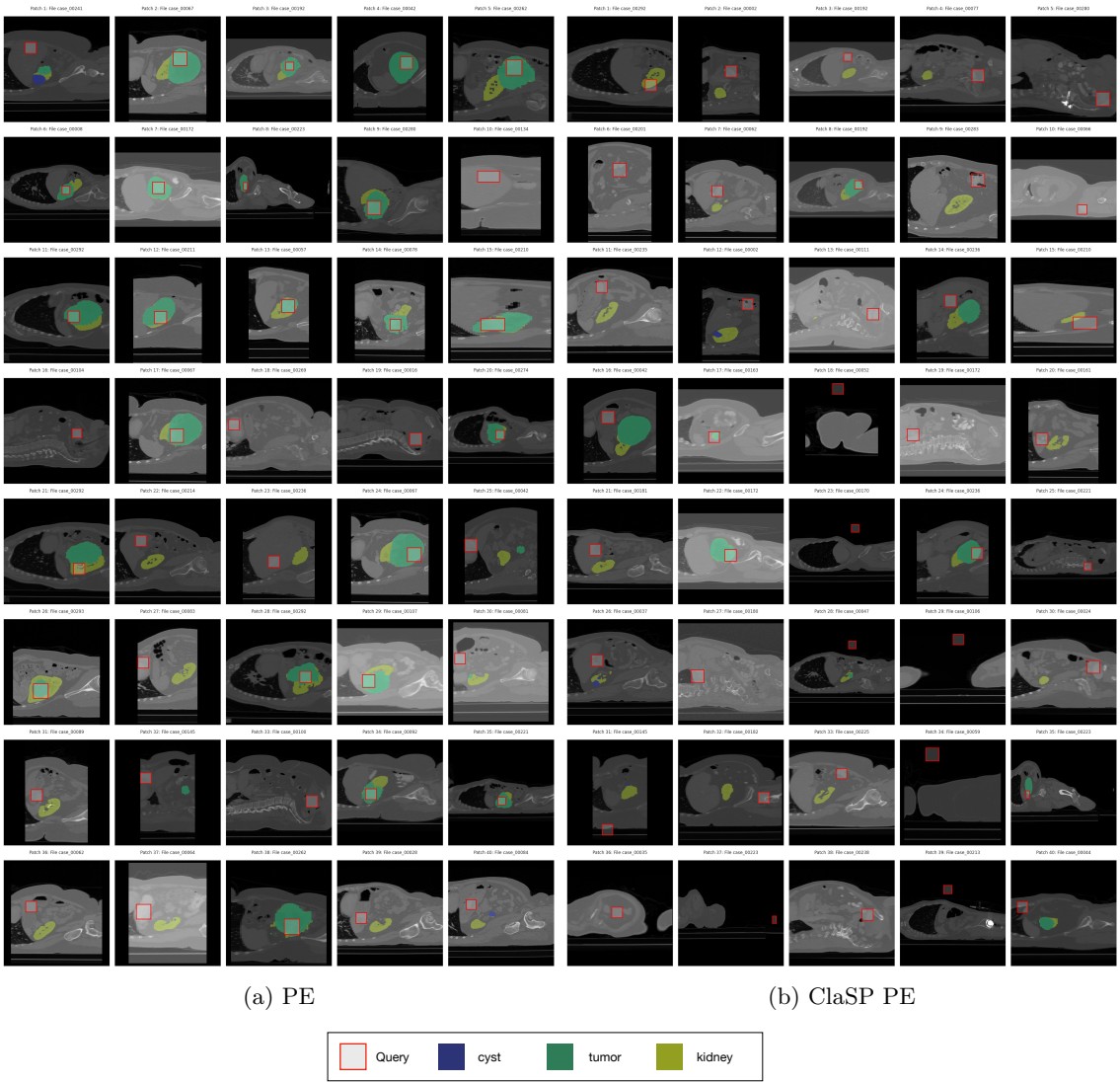

(a) PE                    (b) ClaSP PE

| Query | cyst | tumor | kidney |

Figure 19: Exemplary visualization of the queried patches using PE (a) and ClaSP PE (b) after the first AL loop on the KiTS dataset (same seed to ensure comparability). For 2D visualization, we selected the center slice of the 3D patches. Best viewed zoomed in.

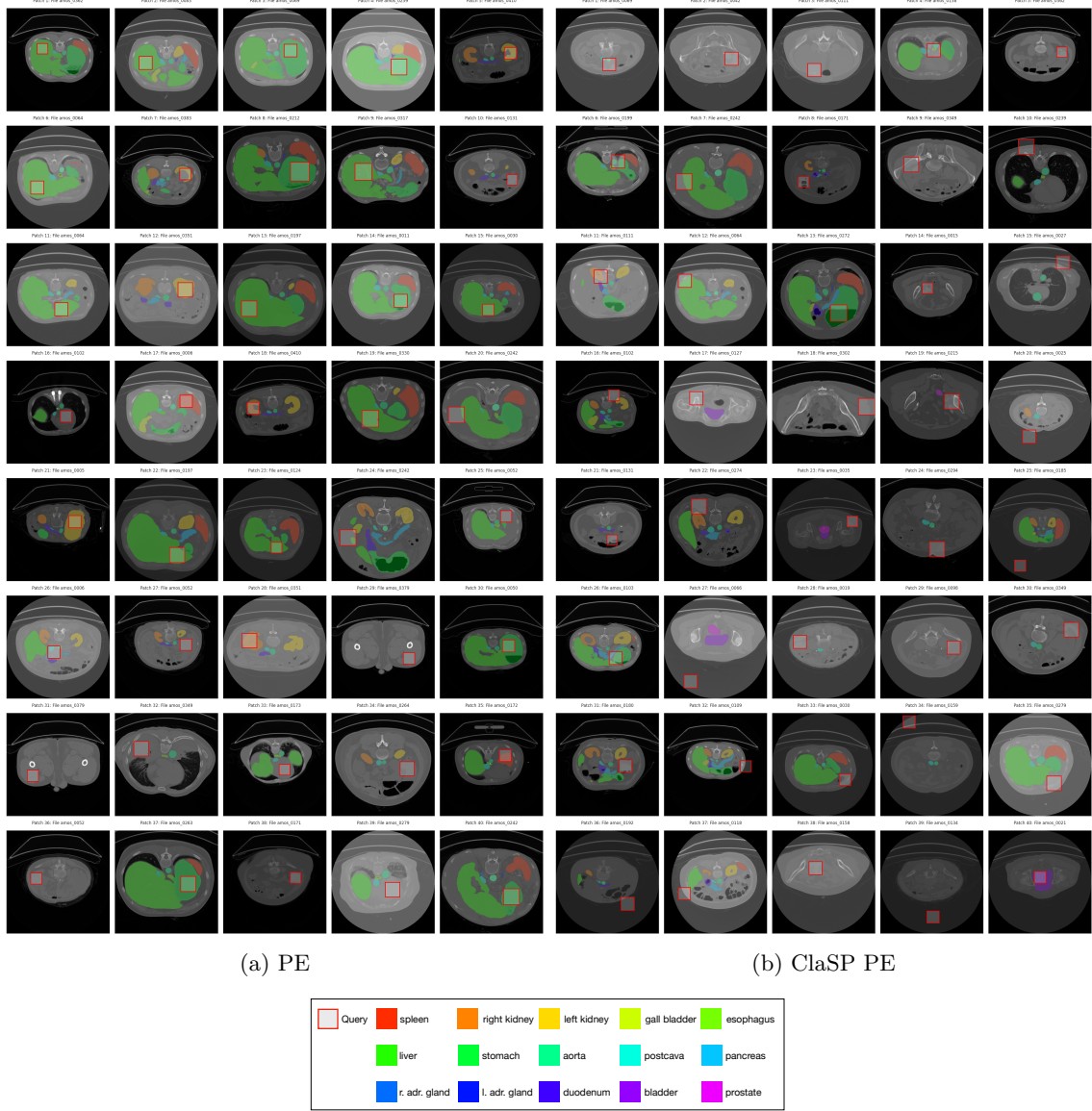

(a) PE                    (b) ClaSP PE

Figure 20: Exemplary visualization of the queried patches using PE (a) and ClaSP PE (b) after the first AL loop on the AMOS dataset (same seed to ensure comparability). For 2D visualization, we selected the center slice of the 3D patches. Best viewed zoomed in.

## I.2    Stratification Visualization

Figure 21 illustrates the class-wise stratification defined in eq. (3), based on predictive entropy. For this example, we use models from the first loop of the Low-label regime in the main nnActive benchmark setting.

The figure shows that this stratification shifts the regions of high uncertainty for each class toward the areas where the model's predictions indicate these classes are present.

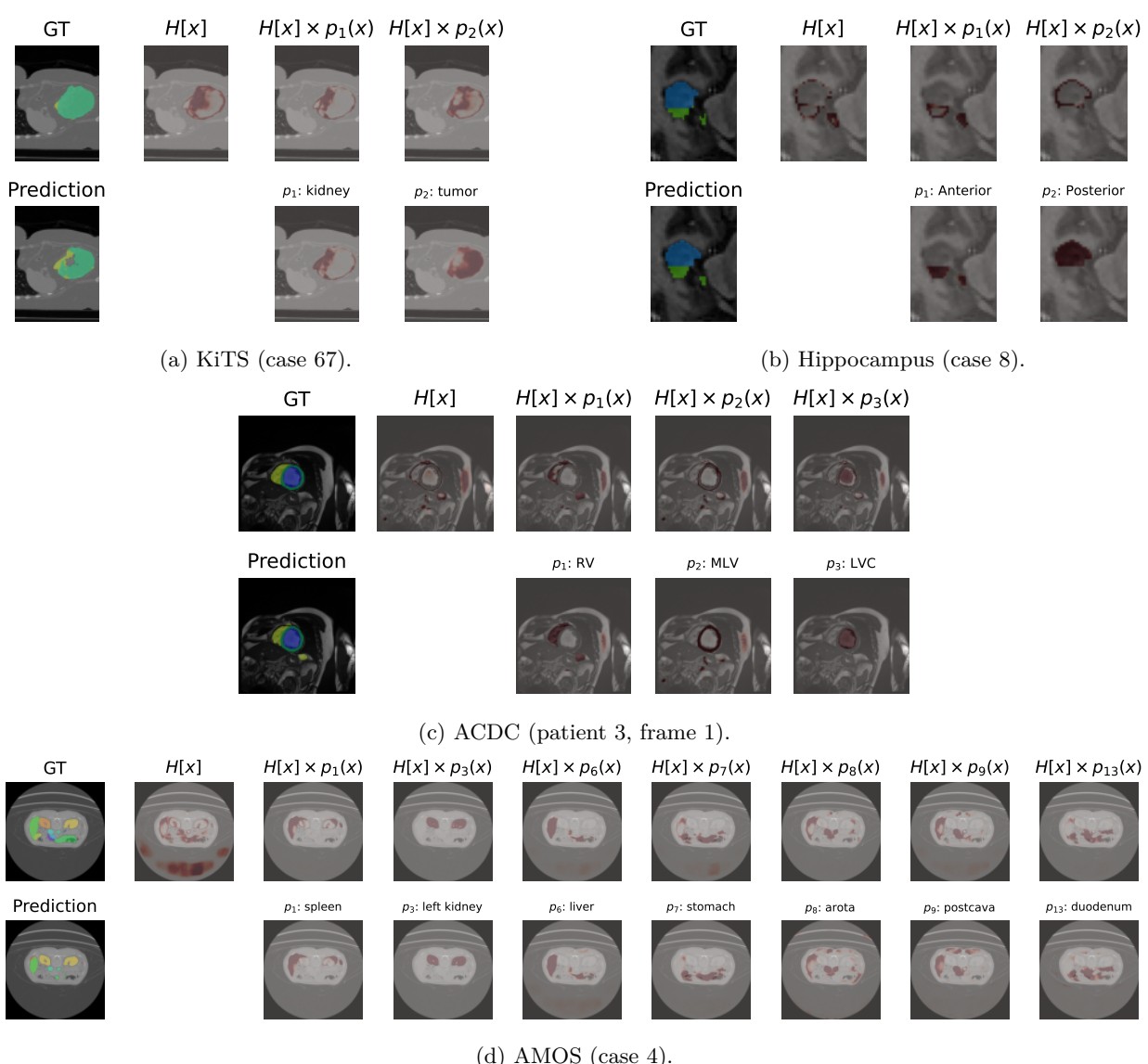

(a) KiTS (case 67).

(b) Hippocampus (case 8).

(c) ACDC (patient 3, frame 1).

(d) AMOS (case 4).

Figure 21: Visualizations of the stratification mechanism of ClaSP PE for exemplary cases of each nnActive benchmark dataset. For each predicted class in the displayed slice, the class probabilities $p_i(x)$ as well as the weighted entropy maps $H[x] \times p_i(x)$ are shown, which lay the basis of the stratified querying. The colormaps are rescaled for each individual image. To avoid outliers distorting the color mapping, we clip high values at the 98% quantile of the data.

