# OpenReview forum: "Finally Outshining the Random Baseline: A Simple and Effective Solution for Active Learning in 3D Biomedical Imaging"
_TMLR — Accepted by TMLR_

### Review · Reviewer_6yXK · 2025-10-30

**Summary Of Contributions:**

The authors propose a patch-based active learning method for 3D medical segmentation which leverages uncertainty-based scoring, class stratified sampling, and an exponential scheduler with robust improvement over random baselines for several different evaluation settings.

**Additional Comments:**

N/A

**Audience:**

Yes

**Audience Explanation:**

Yes, the robust evaluation of this method, as well as others, would be very interesting to the medical imaging community.

**Claims And Evidence:**

No

**Claims Explanation:**

The first main contribution of the paper is that "we propose ClaSP PE, a simple and effective query method that systematically addresses key limitations of current uncertainty-based AL methods."

In general, the method seems straightforward and reasonable. The paper argues that class-stratified sampling has not been done before and addresses key limitations of entropy-based methods. However, class-stratified sampling has not been clearly defined—how is the sampling done precisely? Would the goal be that the sum of the probabilities of the different classes is roughly equal after obtaining all patches? If the goal is class balancing, I would argue that this approach is beneficial because the metric used is the mean DICE (the average DICE score across all classes). The method may not generalize to other evaluation metrics. This should be clearly emphasized in the paper. While taking the average performance over all classes is common, there are situations in which this may not be the ideal solution.

Additionally, while the paper argues that their method is hyperparameter-free and robust to overfitting, it provides few details about the hyperparameters used for the exponential scheduler. Please provide more information about this and indicate whether any of it was changed across different evaluation settings.

The second main claim of the paper is that "we conduct a large-scale evaluation, demonstrating that ClaSP PE is the first AL method to bring reliable performance improvements over standard and improved random sampling baselines for 3D biomedical image segmentation in a close-to-production environment."

Overall, I appreciate the choice of metrics: final DICE score, AUBC, FG-Eff, and PPM all seem reasonable choices that generally reflect the main goals of active learning. Additionally, I generally appreciate the choice of entropy methods. While the inclusion of diversity-based methods would be nice, as you mentioned, applying them with pairwise comparisons on the patches is computationally intensive.

While the method generally improves on the baselines, I believe the relatively lower performance on FG-efficiency warrants further discussion. You state, "it is also among the best-performing methods measured by FG-Eff," which dismisses the issue instead.

The discussion on the failure case on the AMOS dataset was appreciated. The discussion that more complex cases require longer training is appreciated for real-world application. Additionally, the ablation was good, but ideally, there could be some ablation results related to the scheduler.

The rollout performance is good. The colors in Table 1 are hard to read - it would be better if the differences were more clearly visible. It should also be easier to compare performance between the first and second settings to see how much the performance suffered in the real-world setting, if at all (to me, the rollout performance may have been comparatively worse, but it's hard to tell).

I would be wary of the tone of statements like this: "Importantly, the partially competitive performance of PE does not undermine our conclusions" because it is suggestive of bias, as it assumes there is a competition between your method and this one, in which case, you would obviously be biased towards your method. You then make a statement "the results on the nnActive benchmark (fig. 2)..."; however, as mentioned, the hyperparameters for your particular method may be fine-tuned to that evaluation task. In general, this paragraph could be written in a more even-handed manner. The following paragraph as well could be written in a more even-handed manner and use less bias ("also outperforming Predictive Entropy significantly in 25% of all cases while only being outperformed in 5%").

The third main contribution of the paper is that "we provide practical Guidelines for Real-World Deployment to configure AL pipelines on new datasets, facilitating straightforward adoption of ClaSP PE." However, this is not discussed in detail in the main paper and would arguably not be a main contribution of this work.

**Requested Changes:**

Please address the concerns above. Additional context, as discussed above, would be helpful to the research community.  Additionally, while the paper discussed slice-based and patch-based active learning, there are other options, including scribbles, points, superpixels, text, etc. Several of these annotation methods are less expensive than full annotations and may be less expensive than patches. Also, the annotation complexity can vary across patches. While a comprehensive experimental comparison is obviously not feasible, a good discussion on the pros/cons of the different choices would be beneficial to researchers and help others contextualize the contributions of your work.

Some minor comments. Please consider reorganizing the method section. Ideally, provide the whole method overview first, and then give each of the sub-components, rather than each of the components, and then the full method. It wasn't very clear on the first read.

Another minor comment: on page 8, you forgot to include a percent sign with the alpha value

---

> ### Author Response · Authors · 2025-11-27
> **Rebuttal**
>
> Thank you again for your valuable comments, and for taking the time to read our general reply, as well as considering our point-by-point comments here:
>
> ---
>
> W1. “class-stratified sampling has not been clearly defined [...]”
> * Thank you for bringing up this point. The goal of our class-stratified sampling strategy is to obtain an equal number of patches per predicted class by selecting the top $N_c$ patches per class $c$. In the revised manuscript, we updated the corresponding method section to clarify the sampling procedure.
>
> ---
>
> W2. “The method may not generalize to other evaluation metrics.”
> * We agree that this should be clearly stated and added this point to our limitation section to ensure our work is interpreted in the correct context.
>
> ---
>
> W3. “provides few details about the hyperparameters used for the exponential scheduler”
> * In the spirit of providing a simple yet effective method, we fixed the hyperparameters (initial and final noising strength) across all evaluation settings and performed no finetuning. For improved clarity, we included this information as well as the specific scheduling in the main paper in the method section.
>
> ---
>
> W4. “the relatively lower performance on FG-efficiency warrants further discussion”
> * Thank you for raising this point. In the revised version, we added a more nuanced discussion of the interplay of segmentation performance and annotation efficiency in the results section.
> * Updated manuscript: “although ClaSP PE does not always achieve top FG-Eff, it consistently ranks among the most efficient methods. This reflects an inherent interplay between segmentation performance and annotation efficiency, where methods that strongly focus on highly informative regions can improve Dice scores but may risk inefficient use of annotated foreground”
>
> ---
>
> W5. “ideally, there could be some ablation results related to the scheduler”
> * We performed single prototyping runs with a linear, sigmoid and exponential scheduler where the $\beta$-parameter is interpolated from 1 to 100 during prototyping and did not find any meaningful differences. Therefore, we decided to use an exponential scheduler.
> * We agree that more experiments might be useful here. To give clear guidance with regard to the scheduler, we deem a full evaluation on the nnActive Benchmark (Main and Patchx1/2) to be necessary which requires over 6,6k GPU hours (A100) or at least the Main setting requiring half of it. Based on the little effect we observed during prototyping, we spent our compute budget on other ablations.
>
> ---
>
> W6. “tone of statements like [...] is suggestive of bias”
> * Thank you for raising this concern. In response to your feedback, we reformulated the two mentioned paragraphs in a more descriptive tone.
>
> ---
>
> W7. “[Guidelines for Real-World Deployment] would arguably not be a main contribution of this work”
> * We agree that the Guidelines would not be a main contribution given that there is no detailed discussion in the main paper. Therefore, we removed it from the list of main contributions in the introduction, and instead focus on presenting the evidence based on the Roll-Out study which relies on these Guidelines.
>
> ---
>
> RC1. “while the paper discussed slice-based and patch-based active learning, there are other options, including scribbles, points, superpixels, text, etc”
> * We added a discussion to our limitations section in the main paper (“On the Importance of Query Design and Annotation Technique”) where we discuss the importance of the query design and annotation technique to ensure that our results are interpreted in the correct context for 3D patch query design which can be combined with further annotation techniques, including scribbles, points and superpixels.
>
> ---
>
> RC2. “Please consider reorganizing the method section”
> * Following your suggestion, we improved the method description in the revised version. In particular, we introduced the method first, followed by descriptions of its components.
>
> ---
>
> Thank you for your constructive feedback. As we believe to have resolved your comments, please let us know if there are any remaining concerns that would hinder a recommendation for acceptance.

---

> ### Comment · Reviewer_6yXK · 2025-12-19
>
> Thank you for your response. This has addressed my concerns.

---

### Review · Reviewer_Cfj7 · 2025-11-07

**Summary Of Contributions:**

The paper presents a simple active learning method for the annotation-efficient segmentation of 3D medical images. The method, called Class-stratified Scheduled Power Predictive Entropy (ClaSP PE), combines two ideas: 1) stratified class sampling based on prediction pseudo-labels, a power-noising of scores based on (Kirsch et al., 2023) which induces a more uniform random sampling of patches in the early stages of active learning (AL). A comprehensive set of experiments is used to evaluate the proposed method, including a various datasets and baselines. Results show the method's competitive advantage over other AL approaches.

**Additional Comments:**

* The paper uses 3D patches as samples to annotate. While this choice makes sense for 3D segmentation, I am not sure it is optimal for the expert performing the annotation as labelling a 3D patch is harder than labelling a 2D image. It would interesting to discuss this point in the paper.

* Figure 4 is a bit hard to follow, in particular, how the label regimes are considered in the analysis.

* p. 10: "while while winning"

**Audience:**

Yes

**Audience Explanation:**

* Active learning is a crucial challenge in medical imaging, where obtaining annotations is and often requires expert knowledge. The proposed method is simple and effective, and could easily be implemented in current pipelines.

* Beyond introducing the proposed method, the paper provides a comprehensive set of experiments that can serve as a valuable reference for future research in the field.

**Claims And Evidence:**

Yes

**Claims Explanation:**

* Authors acknowledge that their work does not focus methodological novelty but instead on providing
a simple solution built on existing strategies. I agree with this assessment: the power noising approach is adopted from Kirsch et al. (2023), and the class-wise stratified sampling method, while interesting, is a relatively straightforward idea.

* I also concur with authors that random selection remains a hard to beat baseline, in particular when at early stages of AL training, and that having a stratified class sampling can avoid over-selecting larger or less certain classes.

* Regarding the results, although the proposed method performs well in ranking-based analyses, the actual gains in AUBC and final Dice are less evident. While the authors acknowledge this limitation, it raises questions about the method’s practical usefulness.

**Requested Changes:**

* The claim that the proposed method is the first to outperform (improved) random baselines for 3D image segmentation may be overstated. Previous approaches, in particular (Gaillochet et al., 2023), also achieved similar results. While these approaches performed segmentation in 2D, they could be easily adapted to the 3D patch setting of the paper. Adding this approach as baseline would strengthen the claim.

* Some statistical analyses are conducted at a low level of significance (p<0.1). I suggest the authors to try a more common p-value threshold such as p<0.05.

---

> ### Author Response · Authors · 2025-11-27
> **Rebuttal**
>
> Thank you again for your valuable comments, and for taking the time to read our general reply, as well as considering our point-by-point comments here:
>
> ---
>
> W1. “the actual gains in AUBC and final Dice are less evident”
> * Overall the design philosophy for ClaSP PE is about consistent improvements with few losses (none in the best-case scenario). We give more detail to this point in the general response in Issue 2. Generally, we would like to stretch the point that AL methods lead to smaller ”absolute performance improvements” when combined with well optimized models, which is well documented  (Beck 2021, Lüth 2023, Mittal 2019). By using nnU-Nets self-configuring pipeline our models are well adapted to low-budget scenarios.
>
> ---
>
> W2. “I suggest the authors to try a more common p-value threshold such as p<0.05” for the Nemenyi test
> * We changed the p-value to 0.05 in our manuscript. For a more detailed answer please see the general response Issue 1.
>
> ---
>
> RC1: “The paper uses 3D patches as samples to annotate. [...] labelling a 3D patch is harder than labelling a 2D image [...] It would interesting to discuss this point in the paper.”
> * We agree with the reviewer that annotating a single 2D slice can be easier than to annotate a 3D patch. However, this is not necessarily always the case as the annotation of 3D patches allows to focus on smaller regions inside of a 2D image and can be combined with sparse annotation techniques where only a subset of slices is annotated.
> * Based on your feedback, we extended the limitation section with a new paragraph “On the Importance of Query Design and Annotation Techniques” where we discuss this more explicitly, especially with respect to emergent annotation frameworks for 3D biomedical segmentation such as MedSAM (Ma 2024) and nnInteractive (Isensee 2025).
>
> ---
>
> RC2: Overstated claim “first to outperform (improved) random”...
> * We agree with the reviewer and changed our manuscript to give our claims a more detailed scope that focus on our provided evidence. We rephrased this concrete example to: “Within the nnActive framework, we present compelling evidence that an AL method can consistently outperform random baselines adapted to 3D segmentation”.
> * We generally focused on giving a more concrete scope of our claims which we detail in the general response in Issue 3.
>
> ---
>
> RC3: “Previous approaches, in particular (Gaillochet et al., 2023), also achieved similar results.”
> * We appreciate this remark and would like to clarify the relationship between our findings and prior work. First, as mentioned in response to RC2, we downtoned our overall statement to better reflect the scope of our conclusions. Second, Gaillochet et al. (2023) results show that their proposed “Stochastic Batches” improves over top-k sampling by leveraging “the power of randomness during uncertainty-based batch sampling” to increase diversity of queries on two datasets  (MSD Hippocampus and Promise Prostate) when using a limited number of 2D slices which shows a diminishing effect for larger annotation budgets (see Figure 8).
> * Power and Softrank Noising (Kirsch et al., 2023) are also both based on introducing randomness into the query process and also show a very similar general behavior (Kirsch et al., 2023) which also extends to 3D biomedical segmentation by improving diversity which improves performance in earlier stages with diminishing effect in later stages.
> * Given that we extensively benchmark PowerBALD, PowerPE and SoftrankBALD we omit further experiments with AL methods purely introducing stochasticity as (Gaillochet et al., 2023) due to the cost of benchmarking these methods on the nnActive Benchmark (~6.6k GPU hours with an A100) and the extremely similar behavior of these methods.
>
> ---
>
> RC4: Label Regimes in Figure 4 are a bit hard to interpret:
> * We added the following statement to the caption of Figure 4 to describe how the label regimes are considered in the analysis: “Each Label Regime carries 33% of the entire fraction of experiments which is then divided into wins, losses and ties.”
>
> ---
>
> Thank you for your constructive feedback. As we believe to have resolved your comments, please let us know if there are any remaining concerns that would hinder a recommendation for acceptance.
>
> Citations:
>
> (Beck 2021) Beck et al., "Effective evaluation of deep active learning on image classification tasks". arXiv preprint arXiv:2106.15324, 2021.
>
> (Mittal 2019) Mittal et al., "Parting with Illusions about Deep Active Learning". arXiv:1912.05361, December 2019.
>
> (Lüth 2023) Lüth et al., "Navigating the pitfalls of active learning evaluation: A systematic framework for meaningful performance assessment". Advances in Neural Information Processing Systems, 36:9789–9836, 2023.
>
> (Isensee 2025) Isensee et al., "nninteractive: Redefining 3d promptable segmentation". arXiv preprint arXiv:2503.08373, 2025.
>
> (Ma 2024) Ma et al., "Segment anything in medical images". Nature Communications, 15(1):654, 2024a.

---

> > ### Comment · Action_Editor_LP3R · 2025-12-23
> > **Submit final recommendation**
> >
> > Dear reviewer,
> >
> > Can you input your final recommendation?
> >
> > Best, AE

---

### Review · Reviewer_5sG6 · 2025-11-20

**Summary Of Contributions:**

The authors propose a new method for AL for 3D image segmentation for biomedical applications. Their method is based on a simple modification (which is not a flaw) of predictive-entropy sampling. The main motivation is to address two modes of failure of the  uncertainty-based AL in this setting.
1. Imbalance between background and foreground classes and between foreground classes
2. Redundant samples early in the AL cycles that are concentrated close to each other spatially
Authors have used nnunet as a backbone, with the nnactive benchmark. Four seeds per configuration provides some validity to the work.

Also the main claim here is that there method provides the first compelling evidence of an AL method that beats strong baselines in terms of segmentation quality.

**Audience:**

Yes

**Audience Explanation:**

Yes, work is relevant.

**Broader Impact Concerns:**

Nothing in particular to note.

**Claims And Evidence:**

Yes

**Claims Explanation:**

Not fully, but mostly yes, based on the results they presented for the central empirical claim. Also partially, for the bigger general claim.

The main claims/hypotheses they present are:
1. Class‑stratified uncertainty and their scheduling approach to power‑noising can fix the two issues as I wrote above.
2. These improvements will generalize to new datasets when using their approach for patch size/ budgets/schedule.
3. At least for the biomedical segmentation tasks there method provides the strong evidence of consistently beating a strong random baseline.

For 1) the nnactive benchmark is an acceptable choice and the range of ROIs is good (ACDC, AMOS, KiTS, Hippocampus), each with three labels  and 2 query patch sizes. I do wonder how such an algorithm will do with scenarios where you have plenty of very small objects (e.g., lung nodules <5mm). The variants of BALD and random baselines are also acceptable given their focus is on uncertainty-based sampling methods. A lot of tricks used are good practices in the current 3d seg AL literature (patch-based query, partial annotations,  foreground-aware random baselines).
For 2, the generalization study they have used four more datasets with different anatomy and characteristics, and they have configured AL in a somewhat principled manner (e.g., median target size for patch size and budgets proportional to number of classes etc) and achieved without tuning for individual new datasets. This is good practice for the generalization experiment.
For 3, conceptually the main experiments are okay, where we have a nnactive evaluation and a generalization experiment.

Some of my reservations and subdued enthusiasm is due to the fact that by author's own admission in the appendix, the hyperparameter tuning wasn't done rigorously, and it is unclear if random baselines had a fair "compute budget" to be competitive with random baselines. In my own work this is the main problem I have seen before.

In terms of the actual results themselves, fig2 and fig3 support their claim of Clasp PE's superiority compared to random, BALD variants and other PE methods, although it is worth noting that the gains are typically modest (~1 to 2%). On the positive side, these appear to be consistent in direction. With the nnunet backbone which is strong and small budgets, from my experience in non-3d and non-biomedical datasets, I have seen that a few percentage points of improvement can easily lie within the error interval around point estimates, but I have no evidence to say this is what's happening here.

One related interested experiment that the authors do talk about is amos experiment with longer training (200 vs 500 epochs), where the PPM on AMOS shows that clasp-pe’s win/loss ratio vs rrandom 66% fg improves as the models are left to train for longer duration. They note that the "loose" cases are mostly in the low‑label regimes. This supports there argument that some of the residual random‑baseline wins are due to under‑trained models rather than inherent AL limitations, but on the other hand, I would have liked to see a greater emphasis on hyper-parameter tuning at the expense of number of epochs.

The gains are relatively bigger for their generalization study (eg fig6). I liked that they presented these results, even though I think that they could have tried with datasets like LIDC IDRI or similar where you can end up with many small ROIs to segment rather than 1 or few large in a given image.

Somewhat problematic for me are claims like “only method with statistically supported improvements”. This is a statement about average ranks over 24 settings with a fairly loose threshold of statistical significance (p=0.1).

The PPM does show that Random 66% FG still wins in about 20% of settings and the paper acknowledges this but then moves back to making the bigger claims (“generally outperforms random baselines”).

From a statistical viewpoint what the results show is that within nnActive + nnU‑Net + patch‑based AL setup, ClaSP PE is a strong approach, which does appear to be consistently competitive method, with good generalization experimental results to support it.

This part I do believe.

The jump from here to the “first compelling evidence” of general AL superiority in 3D segmentation is an over-reach and should be toned down. I do not blame the authors for using such language which unfortunately I see increasingly as authors struggle to be taken seriously in a review process - I have been there, but let's restore some faith in the review system :)

Paper would also benefit from tighter presentation of language used to describe statistics but I am moving that to the requested changes section.

**Requested Changes:**

Some main points are below

1. Tone down statements like “first compelling evidence” and “first plug‑and‑play AL solution”. Be specific and direct, e.g., “within nnActive with nnU‑Net” or “within patch‑based AL for 3D segmentation”.
2. Be clear in the abstract and conclusion that the evidence is strong for this specific framework and set of datasets. But to be fair and good shepherds of science, let's also be clear that extension to other architectures / training pipelines is a plausible but untested hypothesis.
3. Explicitly describe how many configurations enter each Friedman/Nemenyi test (24) and what a difference in average rank of, say, 0.5 means in terms of effect size.
4. Explain more clearly why a loose threshold of p=0.1 was chosen. On a related point, please do frame the resulting groups as exploratory evidence rather than definitive proof.
5. For PPM, make it explicit that multiple pairwise t‑tests are used without FWER adjustment and that PPM is intended as a descriptive summary. Please do add a short discussion of robustness (e.g., whether main qualitative conclusions persist under a stronger threshold or a non‑parametric test).
6. Figure 2 / 3 and Table 5 can include summary statistics that show the mean difference in Final Dice and AUBC between ClaSP PE and the best improved random baseline, with a 95% CI.
7. In section 6 / appendix H please state that the evidence applies to the nnU‑Net as the backbone, patch‑based querying, and the specific foreground‑aware random baselines; and that it does not directly address other architectures, slice‑based training, or non‑nnActive pipelines. Other studies (including some cited in this manuscript) have shown that changes in architecture can render such results non-applicable. Please be direct in relating this to previous findings that AL conclusions can change with architecture and regularization, to help readers understand the significance of the results within the right context.

---

> ### Author Response · Authors · 2025-11-27
> **Rebuttal**
>
> We wish to thank you for appreciative and honest words on our empirical rigor in evaluation and your valuable comments.
> We discuss the main issues in the general reply and answer your feedback point by point in the following:
>
> ---
>
> Q1. “[...] scenarios where you have plenty of very small objects”
> * We agree that LIDC IDRI would represent an interesting extension of our Roll-out Study. As the breadth of our experiments is considerable, we leave this for future work. Based on our experience, we hypothesize that ClaSP PE shows a similar behavior as on MAMA MIA and Tooth Fairy 2, given shared characteristics (binary classification for MAMA MIA; small query patch to image ratio for Tooth Fairy 2).
>
> ---
>
> W1. “[...] hyperparameter tuning wasn't done rigorously, and it is unclear if random baselines had a fair "compute budget" [...]” & “I would have liked to see a greater emphasis on hyper-parameter tuning [...]”
> * We are not completely sure whether we understand you correctly, therefore please inform us in case we misunderstood this comment.
> * All AL-method hyperparameters, including ClaSP-PE, were fixed across experiments, hence no task-specific tuning was performed which would give an unfair advantage. As for the nnU-Net training hyperparameters, we rely on the nnU-Net self-configuration, known to perform well even for small budgets, as you mention. The only deviation is the reduced number of epochs, which we explicitly evaluate. Of course, further hyperparameter optimization might increase performance further.
>
> ---
>
> W2. “it is worth noting that the gains are typically modest”
> * Thank you for bringing this up. Please find our detailed response in the general response for Issue 1.
>
> ---
>
> W3. “Somewhat problematic for me are claims like [...]” & “The jump from here to the “first compelling evidence” [...] is an over-reach”
> * We agree that our claims require a more specific scope. In response to your feedback, we down-toned these statements and other claims throughout the paper. These exact statements were revised as follows:
>     - “[...] first compelling evidence [...]” to “Within the nnActive framework, we present compelling evidence that an AL method can consistently outperform random baselines adapted to 3D segmentation"
>     - “first plug-and-play AL solution" to “This allows it to be implemented efficiently for use in the 3D biomedical segmentation domain when used inside the nnActive framework"
> * For a more detailed response and further examples, we refer to our general response Issue 3.
>
> ---
>
> RC1. “Tone down statements like [...]”
> * Please see our response to W3.
>
> ---
>
> RC2. “Be clear in the abstract and conclusion that the evidence is strong for this specific framework and set of datasets [...]”
> * Please see our response to W3.
>
> ---
>
> RC3. More detailed description of the Friedman/Nemenyi test
> * Thank you for bringing this up. In the Appendix E.1, we extended the section on the Nemenyi analysis, now providing more details and including a discussion on effect sizes.
>
> ---
>
> RC4. Reasoning behind a loose threshold of $p=0.1$ for the Nemenyi test (fig 2) and treating results more as explorative evidence
> * Based on your feedback we changed the significance level to $p=0.05$ in the revised version of our manuscript. Further, we explicitly state now that “these groups provide exploratory evidence” to ensure they are not seen as definitive proof.
>
> ---
>
> RC5. FWER adjustment for PPM
> * Based on your feedback, we do now explicitly state that our standard PPM does not perform family-wise error rate correction. “These are performed without family-wise error rate correction following (Ash et al. 2020,...)”.
> * Additionally we provide an analysis of the PPM with p-value of 0.02 as well as a Bonferroni corrected version across each single experiment in Appendix E.6.
>
> ---
>
> RC6. “include [...] mean difference in Final Dice and AUBC”
> * Based on your feedback, we now present these summary statistics alongside 95% confidence intervals for all methods in table 1. For a description of the results, we refer to our general response Issue 2.
>
> ---
>
> RC7. clarifying scope and limitations
> * Based on your feedback, we added a detailed list of the limitations mentioned (other architectures, slice‑based training, or non‑nnActive pipelines) to our Limitations Section H and revised our limitations section with the additional point: “Finally, since our empirical evidence is obtained using the nnActive framework with 3D patches as query design, conclusions may differ under meaningful deviations from it, such as alternative segmentation backbones (...) or 2D slice queries.”
> * We also revised our conclusion section accordingly with more explicit pointers to the nnActive framework and 3D patches as query design.
>
> ---
>
> Thank you for your constructive feedback. As we believe to have resolved your comments, please let us know if there are any remaining concerns that would hinder a recommendation for acceptance.

---

> > ### Comment · Reviewer_5sG6 · 2025-12-10
> >
> > Thanks for your response and updating the paper. Many of my concerns have been met, few comments:
> >
> > 1. Given that the table 6 shows mixed results (e.g., for Hippocampus and AMOS, many of the "winners" are not ClaspPE and overall the error bars often show that there isn't a clearly distinguishable difference that would stand the scrutiny of a hypothesis test, with multiple comparisons correction), please provide an interpretation in the main text. I do note that section 4.1 investigates this by training for longer (500 epochs). Can you clarify whether the longer training was afforded to ClaSP-PE and Random both, or only for Clasp PE? Did you also try longer training for Hippocampus, if so, what did you find?
> >
> > 2. Regarding my previous comment on Hyper-parameter selection, you responded by saying that HPs were kept fixed. How were the HPs selected initially, was there any HP sweep performed to finalize and settle on the HPs that were finally used? Can you provide more details?

---

> > > ### Author Response · Authors · 2025-12-11
> > > **Comment**
> > >
> > > Thank you very much for taking the time to read our response and for your follow-up comments:
> > >
> > > ---
> > >
> > > 1.1. “Given that the table 6 shows mixed results [...] please provide an interpretation in the main text”
> > > * First, we want to emphasize that we motivate the ranking-based analysis and the high level of aggregation in the beginning of the results section (section 4). However, we agree that the manuscript would benefit from more details on the results shown in table 6.
> > > * Therefore, we added the following paragraph in the results section (marked blue in the updated paper): “For ACDC and Hippocampus, the absolute performance differences are generally small (table 6) and often fall within the respective error bars. This highlights two important points: (1) broad evaluation across many datasets and label regimes is essential to reveal overall trends, and (2) even when such trends clearly favor a given method, this does not imply that it will yield significant gains over all other methods in every individual scenario.”
> > >
> > > ---
> > >
> > > 1.2. On section 4.1
> > > * All Query Methods were compared when the models were trained for 500 epochs (ClaSP PE, Random 33% and 66% FG). We clarify this in the updated manuscript.
> > > * Our experiment design was mainly guided by the PPM results and that longer training generally improves AL performance on KiTS and AMOS (Lüth et al 2025, Appendix G.2): We only performed 500 epoch experiments for AMOS, because only there we found a clear lack of ClaSP PE performance compared to random (as indicated by the PPM in fig. 9 b) and the loss scenarios on AMOS made up the majority of loss scenarios compared to random. Specifically: Figure 3 shows that ClaSP PE loses against Random 66% FG in 19% of all nnActive benchmark scenarios. At the same time, Figure 9 b shows that 70% of AMOS-specific scenarios are losses against Random 66% FG. Given that there are 4 datasets in total, this corresponds to 17.5% of all nnActive benchmark scenarios. Hence, AMOS makes up most of the loss scenarios.
> > > * As Table 3 shows that longer training substantially improves performance on AMOS and KiTS (full dataset performance, [Test set Mean Dice]) but not so much for Hippocampus and ACDC we assume that the overall results should stay similar with regard to comparing AL methods on the other datasets. For ACDC and Hippocampus due to the fact that even on the entire training set the performance only marginally differs for longer training and for KiTS as ClaSP PE is already performing well.
> > >
> > > ---
> > >
> > > 2. “How were the HPs selected initially?”
> > > * We directly took the Random FG configurations from nnActive.
> > > * For alpha we compared 33% and 66% percent as shown in our ablations following the same values as the FG percentage of the Random FG methods.
> > > * As standard β values for PowerBALD, PowerPE and SoftrankBALD we used 1 as detailed in Kirsch et al. (2023) and following Lüth et al. (2025).
> > > * The initial and final noising strength $\beta_0=1$ and $\beta_{max}=100$ were chosen following the evaluation in Lüth et al. (2025) (Appendix G.3) which parsed a similar range showing that the most crucial factor is a general reduction of beta for larger annotation budgets. Based on this, we kept these parameters fixed as we hypothesized that the value of alpha to be more important for our ablation efforts.

---

> > > > ### Comment · Reviewer_5sG6 · 2025-12-11
> > > >
> > > > Ok, please add the details of your hyperparameter selection in appendix for all your experiments. That satisfies all my concerns.

---

### Author Response · Authors · 2025-11-27
**Rebuttal**

We sincerely thank all reviewers for their valuable comments. The reviewers generally agreed on the added value of our work (“work is relevant” (5sG6), “The proposed method is simple and effective” (Cfj7), “the robust evaluation [...] would be very interesting to the medical imaging community” (6yXK), ”A lot of tricks used are good practices in the current 3d seg AL literature (patch-based query, partial annotations, foreground-aware random baselines)” (5sG6)).

At its core, this work presents ClaSP PE, an AL method which is highly likely to reduce annotation effort for 3D biomedical segmentation within the nnActive framework in as many scenarios as possible, by providing strong empirical evidence. To this end, ClaSP PE directly addresses the highly variable performance of existing uncertainty-based AL methods across datasets and annotation budgets on the nnActive benchmark. An explicit Roll-Out study shows that ClaSP PE generalizes to novel scenarios using our implementation and pre-defined guidelines.

Based on the reviews the following three main issues of our manuscript crystallized which we addressed as follows:

---

**Issue 1: The Nemenyi test was evaluated using a significance value of $p=0.1$** (raised by Cfj7, 5sG6)
* Manuscript: We updated our analysis using the Nemenyi test from $p=0.1$ to $p=0.05$ in Fig. 2, alongside an effect size discussion and general information regarding the Nemenyi test in Appendix E.1.
* Discussion: The overall effects described in our results section, in particular that ClaSP PE lies in the top significance group across all metrics, still hold for the updated significance threshold.

---

**Issue 2: The absolute segmentation performance gains are modest in comparison to other methods.** (raised by Cfj7, 5sG6)
* Discussion: While indeed, the absolute gains when compared to always selecting the best method are limited. The issue lies in the fact that it is not clear which of these other baselines performs best in which evaluation setting. In general, ClaSP PE shows stable performance improvements over most baselines across most of our 24 evaluated settings.
* Manuscript: To support this more explicitly, we added the average AUBC and Final Dice alongside their 95% confidence intervals over the nnActive benchmark (tab. 1), showing that ClaSP PE improves over the best random strategy by 1.67% final dice and 0.48% AUBC.

---

**Issue 3: Claims are formulated in a very general manner.** (raised by 5sG6, Cfj7, 6yXK)

We agree with the reviewers that our claims require more precise wording and a more concrete scope based on our evaluation using the nnActive framework with nnU-Net as segmentation backbone where we query 3D patches, so as to ensure that future work can properly build upon it. Therefore, we downtoned our more general statements across the entire paper by giving them a more concrete scope. Examples include:
* Abstract:  We changed “...first compelling evidence…” to “Within the nnActive framework, we present compelling evidence that an AL method can consistently outperform random baselines”
* Introduction (Contributions): We removed “Guidelines for Real-World Deployment” as a main contribution and instead emphasized the evidence based on the additional Roll-Out study which relies on them.
* Experimental Results on the nnActive Benchmark (Results): multiple overall changes
* Limitations: We added a new paragraph “On the Importance of Query Design and Annotating Technique”, where we explicitly state the importance of these for the overall effective use annotation effort and reinstate our focus on 3D patches as query design during our evaluation alongside its consequences to the extension to 2D slices and potential combination with annotation techniques (e.g. scribbles).

---

We have thoroughly revised our manuscript to address the provided feedback and uploaded the revision, where we **highlight relevant modifications in orange**. Changes include:
* Clearer description of our method by revising the method section and providing additional details directly in the main manuscript.
* Additional analysis of the pairwise-penalty matrix with different p-values in Appendix E.6.

We believe these updates resolve the stated concerns. Please find our point-by-point answers in the respective reviewer sections.

---

### Decision · Action_Editor_LP3R · 2026-01-15

**Recommendation:** Accept as is

**Audience:**

Yes

**Audience Explanation:**

Yes, this work may be interesting for researchers working on medical imaging, and more particularly those whose research interests align with active learning. Nowadays, this group represents an important and growing group within the broader machine learning community.

**Claims And Evidence:**

Yes

**Claims Explanation:**

The core claims of the submission are generally well supported. The authors provide solid empirical evidence for their main contribution, and the revised experiments demonstrate consistent improvements over relevant baselines across multiple datasets. While the gains remain modest, the results are sufficiently robust to substantiate the central argument. Furthermore, despite some broader claims still feel somewhat ambitious or a few concerns not fully resolved, the evidence presented before and after the review process is clear, and adequate to justify the paper’s overall claims.